# Ubiquitination of CLIP-170 family protein restrains polarized growth upon DNA replication stress

Xi Wang [1,5], Fan Zheng [2,5], Yuan-yuan Yi [1], Gao-yuan Wang [1], Li-xin Hong[3], Dannel McCollum [4], Chuanhai Fu [2] ✉, Yamei Wang [1] ✉ & Quan-wen Jin [1] ✉

Microtubules play a crucial role during the establishment and maintenance of cell polarity. In fission yeast cells, the microtubule plus-end tracking proteins (+TIPs) (including the CLIP-170 homologue Tip1) regulate microtubule dynamics and also transport polarity factors to the cell cortex. Here, we show that the E3 ubiquitin ligase Dma1 plays an unexpected role in controlling polarized growth through ubiquitinating Tip1. Dma1 colocalizes with Tip1 to cortical sites at cell ends, and is required for ubiquitination of Tip1. Although the absence of *dma1*+ does not cause apparent polar growth defects in vegetatively growing cells, Dma1-mediated Tip1 ubiquitination is required to restrain polar growth upon DNA replication stress. This mechanism is distinct from the previously recognized calcineurin-dependent inhibition of polarized growth. In this work, we establish a link between Dma1-mediated Tip1 ubiquitination and DNA replication or DNA damage checkpoint-dependent inhibition of polarized growth in fission yeast.

Cell polarization in eukaryotic cells is essential for many cellular and organismal functions. The fission yeast *Schizosaccharomyces pombe* is a rod-shaped unicellular organism that grows by polar extension at both cell poles, and thus represents a unique genetically tractable organism for studying fundamental principles of polarized growth[1]. The activation, maintenance, and inactivation of polarized growth in fission yeast are cell cycle-regulated, and cell division and polar growth seem to antagonize each other, as once cells double their original volume and reach a constant cell length (around 14 μm), they enter mitosis and bipolar cell growth stops until the division is complete and growth then restarts[2] (reviewed in refs. 3,4). Cell polarity is regulated by evolutionarily conserved polarity factors (at least 50 in fission yeast) that function primarily at the cortex of cell ends[4–7]. Among these factors, three conserved microtubule plus-end tracking proteins (+TIPs) including CLIP-170 homolog Tip1, the kinesin-7 family protein Tea2, and the EB1 homolog Mal3 transport the Tea1-Tea4 landmark protein complex along

microtubules to cell tips[8–18]. How polarized growth is coordinated with cell cycle regulation and the role of post-translational modifications of +TIP proteins is still not well understood. One hint was provided by a previous study showing ubiquitination-dependent proteolysis of Tip1[19], though the ubiquitin E3 ligase involved has not been determined.

In this work, we identified the cell end-localized Dma1 as the E3 ubiquitin ligase responsible for Tip1 ubiquitination. This modification on Tip1 is required to restrain polar growth in fission yeast when cells are under DNA replication stress, and this signaling acts in a distinct manner from the previously recognized calcineurin-dependent inhibition of polarized growth.

## Results

### Dma1 physically interacts with Tip1 and localizes at cell ends

Fission yeast Dma1 contains a ring-finger (RF) domain at its C terminus and it has been shown that Dma1 regulates mitotic exit and cytokinesis

[1]State Key Laboratory of Cellular Stress Biology, School of Life Sciences, Faculty of Medicine and Life Sciences, Xiamen University, Xiamen 361102 Fujian, China. [2]School of Life Sciences, University of Science and Technology of China, Hefei 230026 Anhui, China. [3]State Key Laboratory of Cellular Stress Biology, School of Medicine, Faculty of Medicine and Life Sciences, Xiamen University, Xiamen 361102 Fujian, China. [4]Department of Biochemistry and Molecular Pharmacology, University of Massachusetts Medical School, Worcester, MA 01605, USA. [5]These authors contributed equally: Xi Wang, Fan Zheng. ✉e-mail: chuanhai@ustc.edu.cn; wangyamei@xmu.edu.cn; jinquanwen@xmu.edu.cn

by acting as an E3 ubiquitin ligase on the SIN scaffold protein Sid4[20]. Ubiquitination of Sid4 prevents recruitment of the Polo-like kinase Plo1 during a mitotic checkpoint arrest[20]. In order to search for more potential substrates of Dma1 E3 ubiquitin ligase, we performed Dma1 affinity purifications with various versions of Dma1, which either lacked ubiquitin ligase activity due to deletion of the RF domain (Dma1$^{\Delta RF}$) or were fused to lysine-less ubiquitin (i.e. all 7 lysines (7K) mutated to arginines (7R)) with a five tandem glycine linker (5xGly) lying in between (Supplementary Fig. 1a). The rationale for these constructs was that Dma1$^{\Delta RF}$ could bind but not ubiquitinate its substrates, and that the Dma1-5xGly-Ub(7KR) fusions might be self-conjugated to target substrates. To further enrich the potential substrates, we also introduced mutations of *mts3-1* (proteasome subunit) or *dnt1Δ* to slow down the degradation of ubiquitinated proteins or enhance possible target protein ubiquitination, respectively[21,22] (Supplementary Fig. 1b). Interestingly, one major protein, the CLIP-170 homolog Tip1[14], was identified repeatedly (Supplementary Fig. 1b and Supplementary Data 1). The physical interaction between Dma1 and Tip1 was verified by co-immunoprecipitation and pull-down experiments (Fig. 1a, b). The FHA-containing region in Dma1 and the C-terminal half of Tip1 (245-461aa), respectively, were narrowed down as the fragments bridging their interaction (Fig. 1b, c). We noticed that, although Dma1-5xGly-Ub(7KR) fusion failed to self-conjugate to Tip1 as expected, it indeed co-immunoprecipitated significantly more Tip1 than unmodified Dma1 did for unknown reasons (Supplementary Fig. 1c). Nevertheless, this explains why Tip1 was enriched in our series of TAP-Dma1 purifications.

In previous studies, Dma1-GFP or Dma1-mNeonGreen has been observed most intensively at SPB and cell division site[20,23,24]. To gain further insight into the Dma1 connection with Tip1, we closely scrutinized the possible subcellular localization of Dma1-mNeonGreen in regions other than SPB and cell division site. Indeed, we could observe faint but clearly visible punctate Dma1-mNeonGreen signals at cell ends (Fig. 1d). Those signals were more easily discernable in separating cells, newly separated small cells and in elongating cells before early anaphase when more Dma1 started to accumulate at cell division site (Fig. 1d). Interestingly, Dma1-mNeonGreen displayed similar localization pattern at cell ends with Ags1-RFP along the cell cycle, which labeled sites of actively assembled α-(1-3)glucan in cell wall and septum[25] (Fig. 1d and enlarged images in Supplementary Fig. 2), indicating Dma1 is enriched specifically at actively growing cell ends. It has been previously reported that Tip1 localizes at the microtubule tips and cell tips as dots[14]. Intriguingly, punctate Dma1-mNeonGreen signals occasionally colocalized with Tip1-tdTomato at cell ends (Fig. 1e). These data suggested that Dma1 may regulate Tip1 directly at the actively growing cell ends.

## Ubiquitination of Tip1 is mediated by Dma1 in vivo

The RF domain in Dma1 confers E3 ubiquitin ligase activity towards Sid4[20]. To test the possibility that Dma1 is involved in ubiquitination of Tip1, Tip1-6HA was detected by western blotting in immuneprecipitates enriched by tagged ubiquitin and it showed multiple high molecular weight bands, which were abolished in *dma1Δ* cells (Fig. 2a). This is consistent with the earlier study showing that Tip1 is likely ubiquitinated[19]. When cells arrested at different stages in the cell cycle using a drug (hydroxyurea, HU) (S-phase arrest) or various temperature-sensitive mutants (*cdc10-v50* G1 arrest, *cdc25-22* G2 arrest, or *nda3-KM311* prometaphase arrest) were examined, those lower mobility Tip1 bands were detected most strongly in HU-treated cells (Fig. 2b), suggesting that Tip1 is probably strongly ubiquitinated during S phase when the septum is forming or when the DNA replication checkpoint is activated (see below). Furthermore, when Tip1-6HA was immunoprecipitated from HU-treated cells, it also revealed that the high molecular weight bands were largely abrogated by a deubiquitinating enzyme (DUB) USP2, which shows activity towards

multiple types of ubiquitin chains[26], but not by calf-intestinal alkaline phosphatase (CIAP) treatment (Fig. 2c and Supplementary Fig. 3), ruling out that these bands represented phosphorylated Tip1. Also, we found that both *wild-type* and *dma1Δ* cells were well arrested in S phase by HU, therefore the significantly reduced ubiquitination of Tip1 in HU-treated *dma1Δ* cells was not due to failed S phase arrest in these mutant cells (Supplementary Fig. 4). Thus, we conclude that Dma1 is required for Tip1 ubiquitination, especially upon DNA replication arrest.

To further assess the role of Dma1 in Tip1 ubiquitination, we examined whether Dma1 mutants with compromised Tip1 binding capability (R64A mutation) (Fig. 1b) or E3 ligase activity (I194A mutation)[20] would eliminate Tip1 ubiquitination. The *mts3-1* mutation was used to block the degradation of ubiquitinated Tip1 proteins. Strikingly, Tip1 ubiquitination was abolished in both *dma1* mutants (Fig. 2d), demonstrating that both functional FHA and RF domains in Dma1 are required for Tip1 ubiquitination in vivo.

Extensive studies have well established that the fate of polyubiquitinated proteins can be quite different depending on the ubiquitin linkage types, in which Lys48-linked chains target proteins for proteasomal degradation but Lys63-linked chains typically have nonproteolytic functions in cell signaling (reviewed in refs. [27], [28]). The in vitro studies indicated that Dma1 forms polyubiquitin chains on Sid4 passed from E2 enzyme complex, Ubc13-Uev1a, which specifically forms K63-linked chains[20]. To test whether Tip1 ubiquitination by Dma1 affects its protein stability or is only involved in degradationindependent signaling, we first examined the protein levels and relative half-life of Tip1 in *dma1* mutants before and after the addition of protein translation inhibitor cycloheximide (CHX), respectively. We found that Tip1 was more stable in *dma1* mutants (Fig. 2e). And we also found that the half-life of Tip1 was unusually long, which was consistent with the previous observation[19], and the rate of Tip1 proteolysis in *dma1Δ*, *dma1(R64A)* and *dma1(I194A)* mutants was reduced when the protein synthesis was blocked (Fig. 2f). However, when we used ubiquitin mutants K48 or K63 (all Lys residues except Lys48 or Lys63 mutated to Arg, respectively) to examine the Dma1-mediated ubiquitin-conjugation of Tip1, we found that Tip1 was more prominently conjugated by K63 linkage than K48 linkage, though Dma1 catalyzed ubiquitin chain formation of both linkages on Tip1 (Fig. 2g). Together, the long half-life of Tip1 and the preponderance of K63 linkages on Tip1 suggest that Dma1 ubiquitination may primarily serve a regulatory function, although Dma1 does seem to promote Tip1 degradation.

## Cell tip-localized Tip1 is ubiquitinated by cell end-targeted Dma1

Based on our observations that Dma1 facilitates Tip1 ubiquitination and these two proteins colocalize at cell tips, it is plausible to surmise that cell ends are the most likely loci for Tip1 ubiquitination to occur. To test this notion, we first examined whether the absence of Mal3 or Tea2, two other TIP+ components, affected Tip1 ubiquitination. It has been shown that Tea2 is responsible for moving Tip1 and Tea1 to the microtubule plus end[16], and Mal3 is necessary for the proper association of Tea2 with the microtubule[12,29]. Consistent with previous observations, the association of Tip1 with microtubule plus end was abolished in both *mal3Δ* and *tea2Δ* mutants[15,16] (Fig. 3a). However, Tip1 at cell ends remained largely unaltered in *mal3Δ* but almost completely absent in *tea2Δ* cells (Fig. 3a). Consistent with Tip1 ubiquitination occurring at cell ends, Tip1 ubiquitination in *mal3Δ* was comparable to wild-type cells, but significantly reduced in *tea2Δ* cells (Fig. 3b).

Next, we tested whether the forced accumulation of Dma1 at cell ends by fusing to the polarity factor Mod5[30] could possibly allow more Tip1 to be ubiquitinated (Fig. 3c). The engineered Dma1-CFP-Mod5 fusion strongly localized at cell tips but not to SPBs even when Dma1 carried an R64A mutation, which itself failed to localize at cell ends (Fig. 3c, d and Supplementary Fig. 5). Expression of Mod5-fusion protein with wild-type Dma1 (Dma1-CFP-Mod5), but not with Dma1$^{I194A}$,

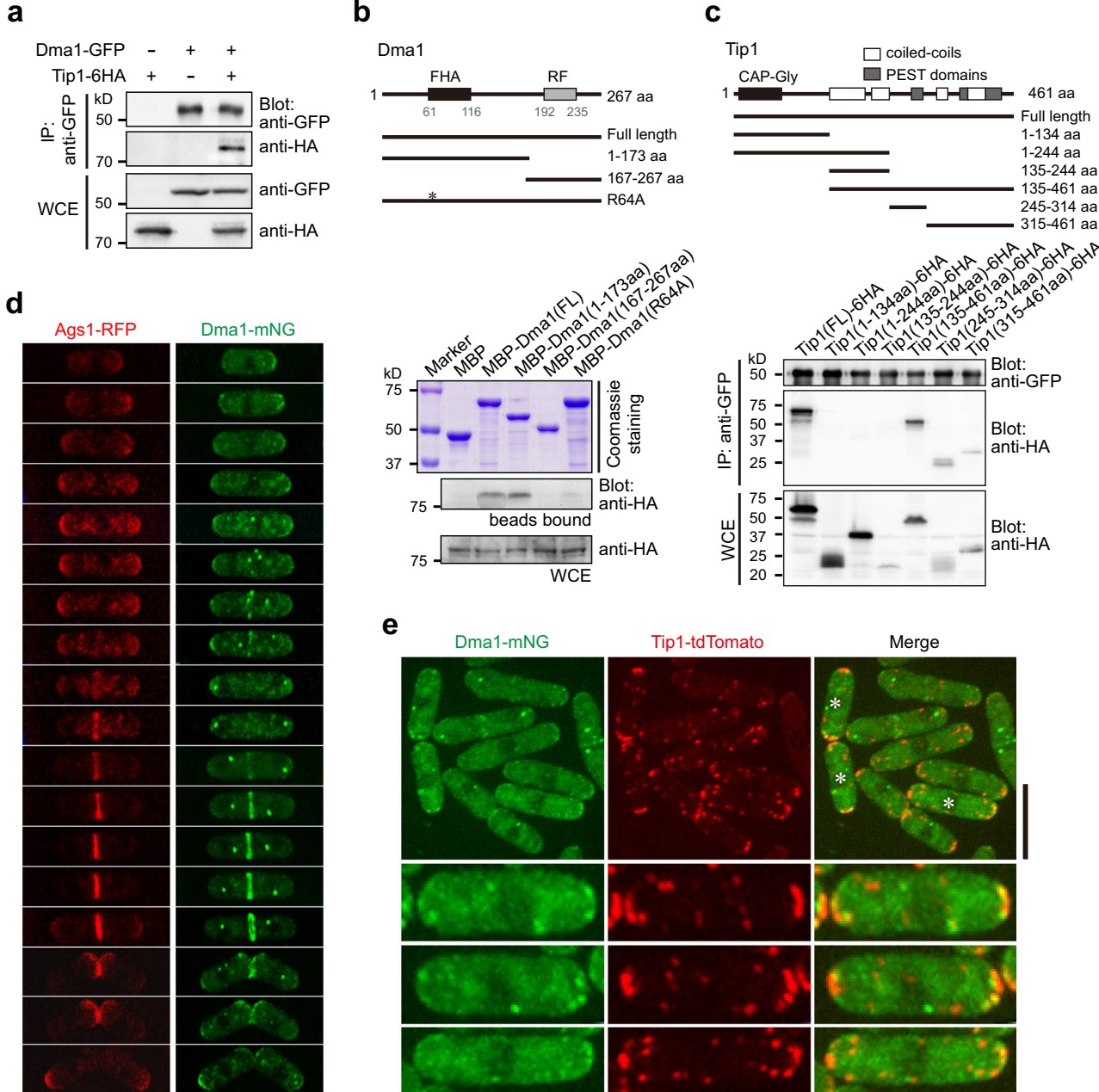

**Fig. 1 | Dma1 physically interacts with Tip1 through its FHA domain and localizes at cell ends during vegetative phases. a** Confirmation of the physical association between Dma1 and Tip1 by co-immunoprecipitation (co-IP). Lysates were prepared from yeast cells expressing either Dma1-GFP or Tip1−6HA, or both Dma1-GFP and Tip1−6HA. Dma1-GFP was immunoprecipitated. **b** Dma1 interacts with Tip1 via its FHA domain in in vitro pull-down assay. Schematic diagrams of bacterially expressed Dma1 fragments or full-length Dma1 with an inactivating mutation (R64A) within FHA domain fused to MBP are shown. Asterisk indicates mutation R64A in Dma1. Yeast lysates were prepared from *tip1-6HA* cells, incubated with bacteria-expressed MBP-Dma1 variants, and subsequently precipitated by amylose resin. **c** Dma1 interacts with the C-terminus fragment of Tip1 covering 245-461aa by co-immunoprecipitation (co-IP). Schematic diagrams of full-length or various truncations of Tip1 tagged with 6HA are shown. Lysates were prepared from *tip1Δ* cells simultaneously expressing GFP-tagged N-terminal fragment of Dma1 (Dma1(1-173aa)-GFP) and HA-tagged full-length or various truncations of Tip1. Dma1(1-173aa)-GFP was immunoprecipitated. **d** Physiological Dma1-mNeonGreen localization throughout the cell cycle. Live cell images were captured in cells expressing both Dma1-mNeonGreen and Ags1-RFP. Enlarged and color merged images are shown in Supplementary Fig. 2. **e** Co-localization of Dma1-mNeonGreen and Tip1-tdTomato at cell ends. Asterisks indicate the representative cells enlarged in insets. Scale bars, 10 μm.

strongly promoted Tip1 ubiquitination (Fig. 3e). It is noteworthy that Dma1^R64A-CFP-Mod5 but not Dma1^R64A-CFP fusion promoted Tip1 ubiquitination to a degree similar to wild-type Dma1-CFP alone, though not as much as Dma1-CFP-Mod5, indicating that targeting Dma1 at cell tips by Mod5 could partially compensate for the loss of binding between Dma1 and Tip1. Our measurement of the Tip1-tdTomato intensity demonstrated that the absence of Dma1 did not affect the

abundance of Tip1 at cell ends upon HU treatment (Fig. 3f), which is consistent with our data showing that Dma1 ubiquitinates Tip1 mainly through K63 linkages (Fig. 2g). Taken all together, although it is clear that Dma1 affects the overall levels and stability of Tip1, the preponderance of K63 linkages on Tip1 suggest the possibility that its ubiquitination by Dma1 may also have a regulatory function at cell ends.

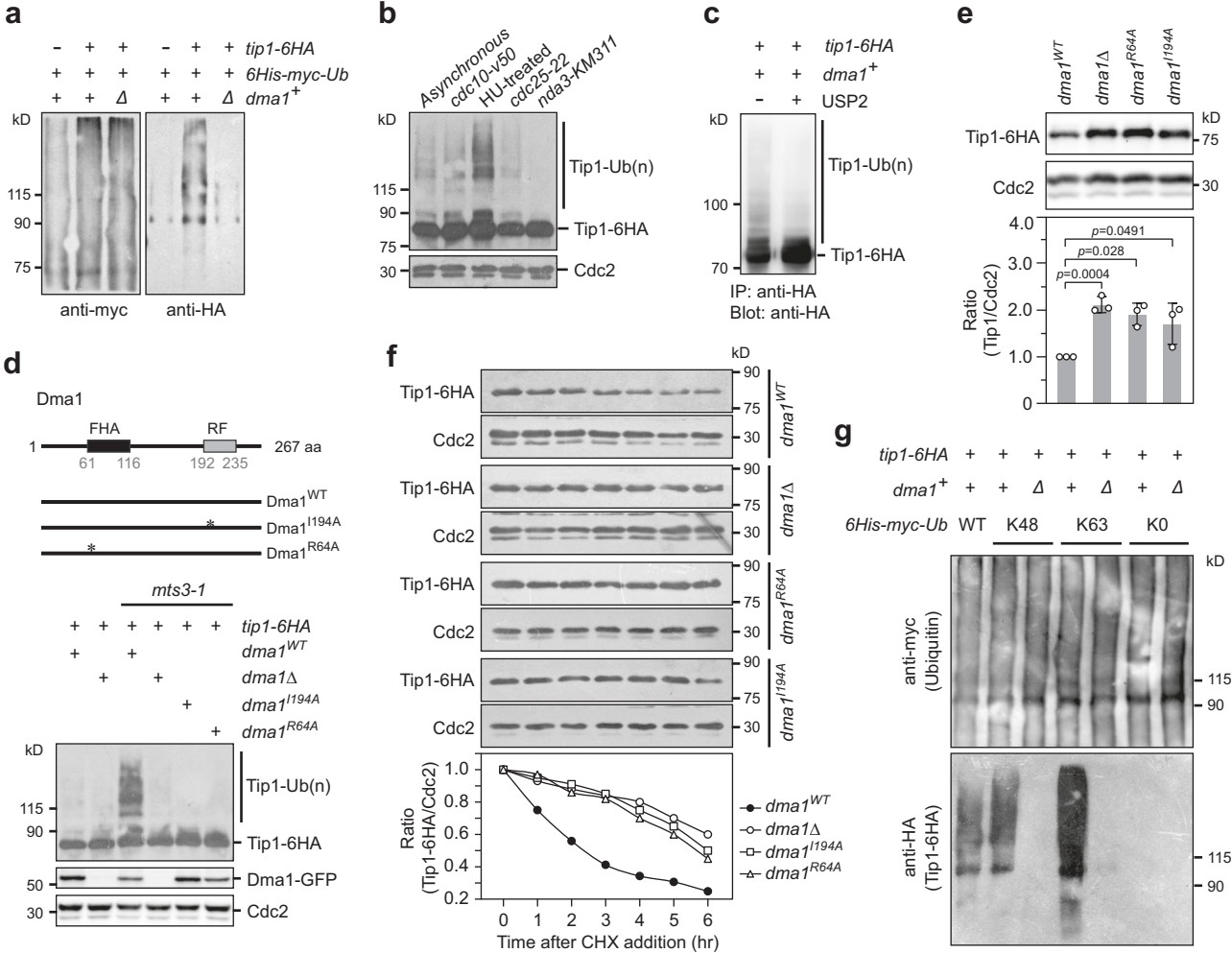

**Fig. 2 | Ubiquitination of Tip1 is mediated by Dma1 in vivo. a** Tip1 ubiquitination is abolished in *dma1Δ* cells. 6His-Myc-tagged ubiquitin (6His-myc-Ub) was expressed from the $P_{nmt1}$ promoter in wild-type or *dma1Δ* cells carrying Tip1–6HA and was pulled down with Ni²⁺-NTA beads. **b** Tip1 is most ubiquitinated in cells treated by hydroxyurea (HU). Yeast cells were arrested at different stages in the cell cycle using a drug HU (12 mM HU for 5.5 h) or various cell cycle mutants and then subjected to immunoblotting with Cdc2 as a loading control. **c** Deubiquitinating enzyme USP2 treatment removes the ladder of slow-migrating bands of Tip1–6HA. Cells grown in YE5S liquid medium were first treated with HU and then cell lysates were prepared and treated without or with USP2 before being subject to immunoprecipitation. **d** Tip1 ubiquitination is abolished in *dma1* mutants. Domain architecture of Dma1 is shown. Asterisks indicate I194A and R64A mutations. Tip1 ubiquitination was detected by immunoblotting in wild-type or *mts3-1* background cells with Cdc2 as a loading control. **e** Tip1 is more stable in *dma1* mutants.

Tip1−HA was detected by immunoblotting in indicated strains with Cdc2 as a loading control. The relative protein level was quantified with a ratio between Tip1-6HA and Cdc2 in wild-type cells being set as 1.00. Bars represent mean ± SD from three independent replicates. Two-tailed unpaired *t*-test was used to derive *p* values. **f** The rate of Tip1 proteolysis in *dma1* mutants is reduced. Time-course samples from cycloheximide (CHX)-treated cells were subject to immunoblotting. Relative Tip1-HA levels were quantified with a ratio between Tip1-6HA and Cdc2 at time 0 h being set as 1.00. Results are representative of two independent experiments. **g** Dma1 mediates ubiquitination of Tip1 via both K48 and K63 chain linkages. Strains expressing wild-type or different mutants of 6His-Myc-tagged ubiquitin as indicated were subjected to ubiquitination assay as in (**a**). The K48 and K63 mutants are ubiquitins with all Lys residues except K48 or K63 mutated to Arg, respectively, and K0 denotes ubiquitin with all 7 Lys residues mutated to Arg.

## Dma1 restrains bipolar growth upon defective DNA replication

Since Dma1 mediates Tip1 ubiquitination, we would expect that the absence of *dma1⁺* could lead to polar growth defects. Surprisingly, we did not observe any significant difference in the percentage of bipolar cells and cell length in vegetatively growing *dma1Δ* mutant and wild-type populations (Fig. 4a, b), or in G₂ phase-arrested *cdc25-22* and *cdc25-22 dma1Δ* cells (Supplementary Fig. 6a, b), when cells undergo a marked transition from monopolar to bipolar growth that is referred to as NETO[2]. However, deletion of *dma1⁺* removed the inhibition of bipolar growth posed by HU treatment and led to increased cell length (Fig. 4a, b), though HU treatment caused equally efficient arrest at S phase in both *wild-type* and *dma1Δ* cells (Supplementary Fig. 4). It has been previously shown that the DNA replication-defective mutant *pol1-1546* undergoes monopolar growth at restrictive temperature[31]. Interestingly, *dma1Δ* also allowed *pol1-1546* to show increased cell

elongation (Fig. 4b), likely due to its switch from monopolar to bipolar growth (Supplementary Fig. 7). These data suggested that Dma1 is required to inhibit untimely bipolar growth upon DNA damage or replication defects.

We noticed that cells with defective DNA replication caused by HU treatment or *pol1-1546* mutation promoted the accumulation of Dma1 at cell ends (Fig. 4c and Supplementary Fig. 8), but not in cells arrested at G₂ phase (Supplementary Fig. 6c), this is consistent with our above result that Tip1 ubiquitination became more evident when wild-type cells were treated with HU (Fig. 2b). To further investigate how Dma1 is actively involved in polar growth delay on perturbation of DNA replication, we monitored the localization of some cell-polarity landmarks, such as Cdc42 and Tea4. The Rho family GTPase Cdc42 polarity module is one of the key effectors and regulators of polarized growth[32], and fluorescent-protein fusions with CRIB (Cdc42/Rac

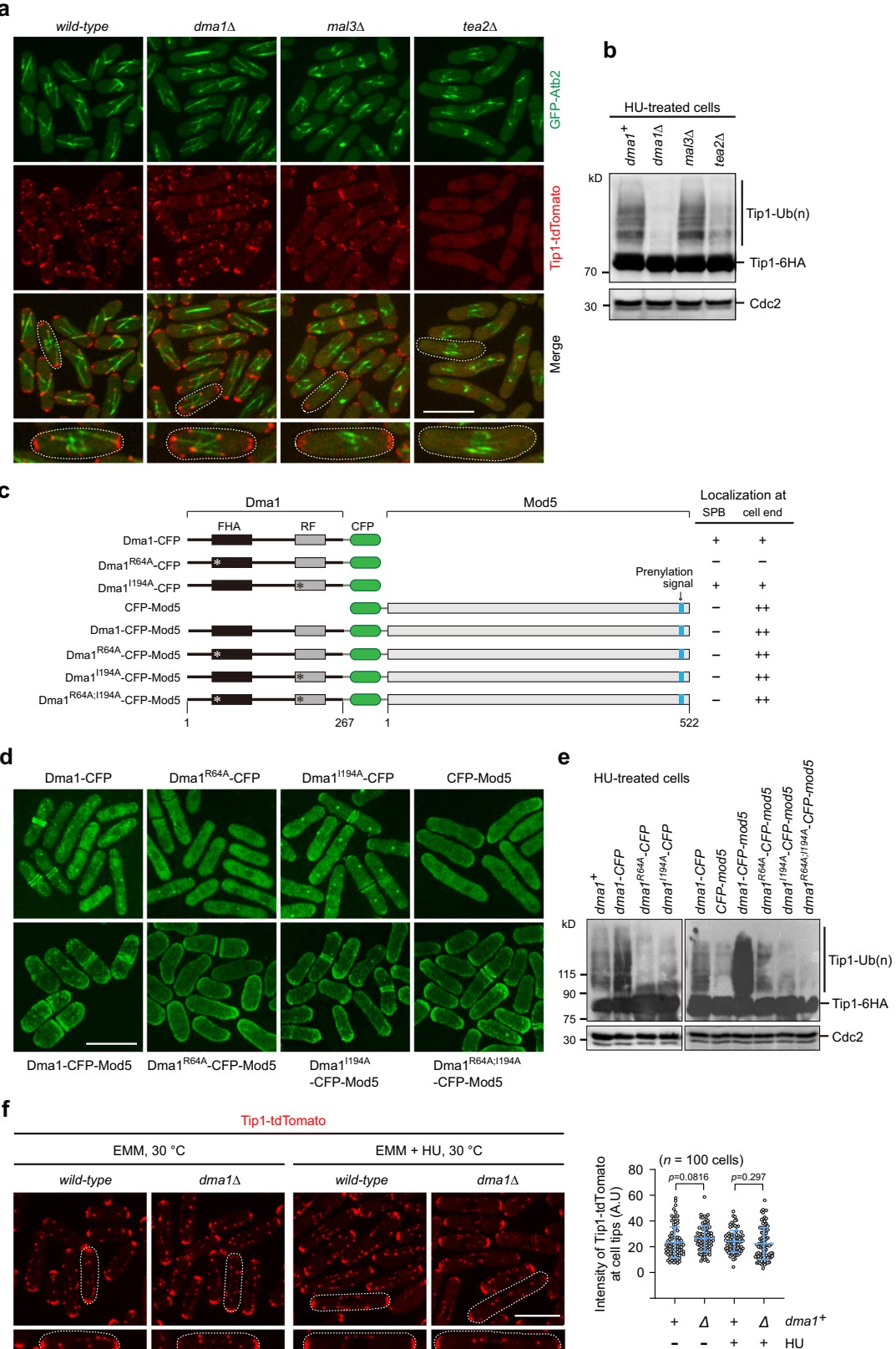

interactive binding motif)-containing domains of Cdc42 are widely used as reporters of active (GTP-bound) Cdc42 localization in vivo[33]. Tea4 is the major linking factor between microtubule- and actin cable-mediated establishment of cell polarity[9]. Strikingly, upon HU treatment, deletion of *dma1+* led to the enrichment of not only active Cdc42 but also Tea4 at cell ends, whereas the overall intracellular levels of these proteins remained unaltered (Fig. 4d, e and Supplementary Figs. 9 and 10). More interestingly, in the presence of HU, the binding between Tea4 and Tip1 was elevated in *dma1Δ* cells and in wild-type cells treated with deubiquitinating enzyme USP2 to remove Tip1 ubiquitination (Fig. 4f), this is consistent with increased polar growth in *dma1Δ* cells (Fig. 4b).

**Fig. 3 | Cell tip-localized Tip1 is ubiquitinated by cell end-targeted Dma1.**
**a, b** Disruption of Tip1 localization at cell ends compromises its ubiquitination. Tip1-tdTomato localization at cell ends was intact in *dma1Δ* or *mal3Δ* but abolished in *tea2Δ* mutant (**a**), and Tip1 ubiquitination was removed or compromised in *dma1Δ* or *tea2Δ* mutant (**b**). Note that Tip1 at microtubule plus ends but not at cell ends was removed in *mal3Δ* cells, and deletion of *tea2⁺* removed Tip1 from both microtubule and cell ends. Representative cells are outlined and enlarged in insets. **c–e** Forced localization of Dma1 at cell ends enhances Tip1 ubiquitination. **c** Schematic diagrams of constructs (not on scale) of Dma1 variants fused to CFP and Mod5 expressed in *dma1Δ* background. Mutations of Dma1 in FHA domain (R64A) or RF domain (I194A) are indicated by asterisks. Localization of fusion proteins at SPB and cell end is summarized. **d** Representative images of Dma1

fusion proteins visualized by CFP fluorescence in live cells. **e** Tip1 ubiquitination is remarkably enhanced in cells expressing Dma1-CFP-Mod5-fusion proteins. Tip1 ubiquitination was detected by immunoblotting in cells treated with HU. Cdc2 was used as a loading control. **f** Abundance of Tip1 at cell ends is not affected by Dma1. Images of Tip1-tdTomato visualized in wild-type and *dma1Δ* live cells treated without or with HU and the quantified fluorescence intensities of Tip1-tdTomato at cell ends are shown. Representative cells are outlined and enlarged in insets. Measured intensities from each individual cell tip are represented by circles. Pooled data from three independent experiments are shown as mean ± SD; *n* = 100 cells analyzed in total for each strain; *p* values were determined by a two-tailed unpaired *t*-test. A.U. arbitrary units. Scale bars, 10 μm.

## Targeted deubiquitination of Tip1 promotes cell polar growth

Our above results suggested that modification of Tip1 by Dma1-mediated ubiquitination is most likely responsible for polar growth inhibition upon DNA replication stress. To further test this possibility, we examined whether deubiquitination of Tip1 affects polar growth by using approaches involving an in cis and an in trans deubiquitinating enzyme (DUB)-fusion, respectively. First, by adopting an established method[34,35], targeted deubiquitination of Tip1 was achieved by its fusing with the catalytic domain of deubiquitinating enzyme UL36 (Fig. 5a). The Tip1-UL36 fusion but not its catalytically DUB-defective counterpart Tip1-UL36(C40S) suppressed Tip1 hyper-ubiquitination upon HU treatment (Fig. 5b). Accordingly, the presence of Tip1-UL36 but not Tip1-UL36(C40S) led to higher proportion of cells with bipolar growth and increased cell length (Fig. 5c and Supplementary Fig. 11a). As an alternative strategy to achieve targeted deubiquitination of Tip1, Tip1 was fused with the PYL domain (residues 33-209) of the regulatory component of abscisic acid (ABA) receptor PYL1 and the catalytic domain (residues 201-875) of fission yeast deubiquitinating enzyme Ubp7 was fused with the ABI domain (residues 126-423) of ABI1 (Fig. 5d). In the presence of the plant hormone ABA, the PYL and ABI domains are sufficient to form a tight complex[36,37], which should promote the formation of complexes of Tip1-PYL and Ubp7-ABI and their efficient recruitment to cell ends (Fig. 5e, f). As expected, ABA-induced in trans proximity between Ubp7 and Tip1 suppressed Tip1 hyper-ubiquitination and led to increased bipolar growth and cell length to a similar degree to *dma1Δ* cells (Fig. 5g, h and Supplementary Fig. 11f). These effects required the DUB enzymatic activity of Ubp7, as the presence of ABI fusion with the enzymatically inactive version of Ubp7 carrying C217S mutation[34] failed to deubiquitinate Tip1 or promote bipolar growth upon HU treatment (Supplementary Fig. 11b–f). These data firmly supported the idea that the unrestrained bipolar growth in *dma1Δ* cells is a consequence of loss of Tip1 ubiquitination which leads to the accumulation of major polar growth-promoting factors at cell ends.

## Dma1 cooperates with calcineurin in inhibition of polar growth

One previous study has established a link between the type 2B protein phosphatase calcineurin-mediated Tip1 dephosphorylation and +TIPs-dependent NETO delay in the presence of DNA replication stress[31]. To examine whether Dma1 functions together or in parallel with calcineurin in inhibition of polar growth, we first tested whether calcineurin is involved in Tip1 ubiquitination. We found that Tip1 was efficiently ubiquitinated in the absence of the calcineurin catalytic subunit Ppb1, with a similar degree of modification to that of wild-type cells (Fig. 6a). We also measured cell length of *pol1-1546 dma1Δ ppb1Δ* cells, and found that *dma1Δ* caused an additive effect to deletion of *ppb1Δ* on cell length at both permissive and restrictive temperatures (Fig. 6b and Supplementary Fig. 12). Furthermore, viability of the *dma1Δ ppb1Δ* cells was more severely dropped than that of either single mutant (Fig. 6c, d and Supplementary Fig. 13). These results suggest that Dma1 and calcineurin regulate polar growth by distinct modifications of Tip1, and both pathways redundantly contribute to the maintenance of viability under the DNA replication checkpoint activation.

## Discussion

In summary, our current study has unveiled an unexpected functional connection between Tip1 ubiquitination and DNA integrity checkpoint-induced polarized growth delay in fission yeast (Fig. 6e). As an E3 ubiquitin ligase, Dma1 seems to have multiple substrates for modifications. During a mitotic checkpoint arrest, Dma1 restrains mitotic exit and cytokinesis by targeting SIN scaffold protein Sid4 for ubiquitination and subsequently impeding Plo1 kinase recruitment at SPBs[20]. When fission yeast cells experience DNA replication or damage stresses during S phase, Dma1 mediates ubiquitination of Tip1 specifically at cell ends to set brakes to inhibit untimely bipolar growth (this study). Although it is clear that Dma1 affects the overall levels and stability of Tip1 (Fig. 2e, f), which is similar to that reported by the previous study[19], the preponderance of K63 over K48 linkages on Tip1 (Fig. 2g) suggests the possibility that its ubiquitination by Dma1 may also have a regulatory function. We favor the notion that Dma1-mediated Tip1 ubiquitination likely affects nonproteolytic functions of Tip1 in polar growth by inhibiting its association with Tea4 at cell ends also based on two other important observations. First, the absence of *dma1⁺* does not affect the abundance of Tip1 measured by its intensity at cell ends with or without HU treatment (Fig. 3f), this indicates that the Tip1 ubiquitinated by Dma1 at cell ends is likely not targeted for degradation. Second, deletion of *tip1⁺* does not rescue the increased growth phenotype of HU-treated *dma1Δ* cells (Supplementary Fig. 14), this suggests that Dma1 may be working through regulating Tip1 through K63 ubiquitination, which cannot be mimicked by deletion of Tip1. Thus, fission yeast has evolved at least two parallel mechanisms involving Dma1 and calcineurin phosphatase respectively to minimize energy expenditure to cell growth on perturbation of DNA replication.

Although a functional interaction of their homologs has not yet been observed in higher eukaryotes, our study sheds light on a potentially conserved connection between ubiquitination of Tip1 homolog CLIP-170 and factors involved in cell polarity and directional cell migration in human tumor cells. It has been reported that CLIP-170 is highly expressed in human breast tumor cells and involved in migration of vascular endothelial cells to promote tumor angiogenesis[38], and phosphorylation of CLIP-170 by AMP-activated protein kinase is essential for human epithelial cell polarity and directional cell migration[39]. Dma1 belongs to a small class of proteins carrying both FHA and RF domains and has two homologs in humans, a tumor suppressor CHFR and RNF8[40,41]. The RF domain confers E3 ubiquitin ligase activity to these proteins[20,42]. Very interestingly, both CHFR and RNF8 are also checkpoint proteins, CHFR delays the cell cycle before prophase upon microtubule stress, whereas RNF8 halts the cell cycle at G₂/M transition when double-strand DNA breaks are detected[40,41,43–46] (reviewed in ref. [47]). Thus, CLIP-170, CHFR, and RNF8 all have pathological functions in cancer, and the potential interaction between CLIP-170 ubiquitination and CHFR or RING8 E3 ligases might have clinical implications, such as the therapeutic treatment of cancers.

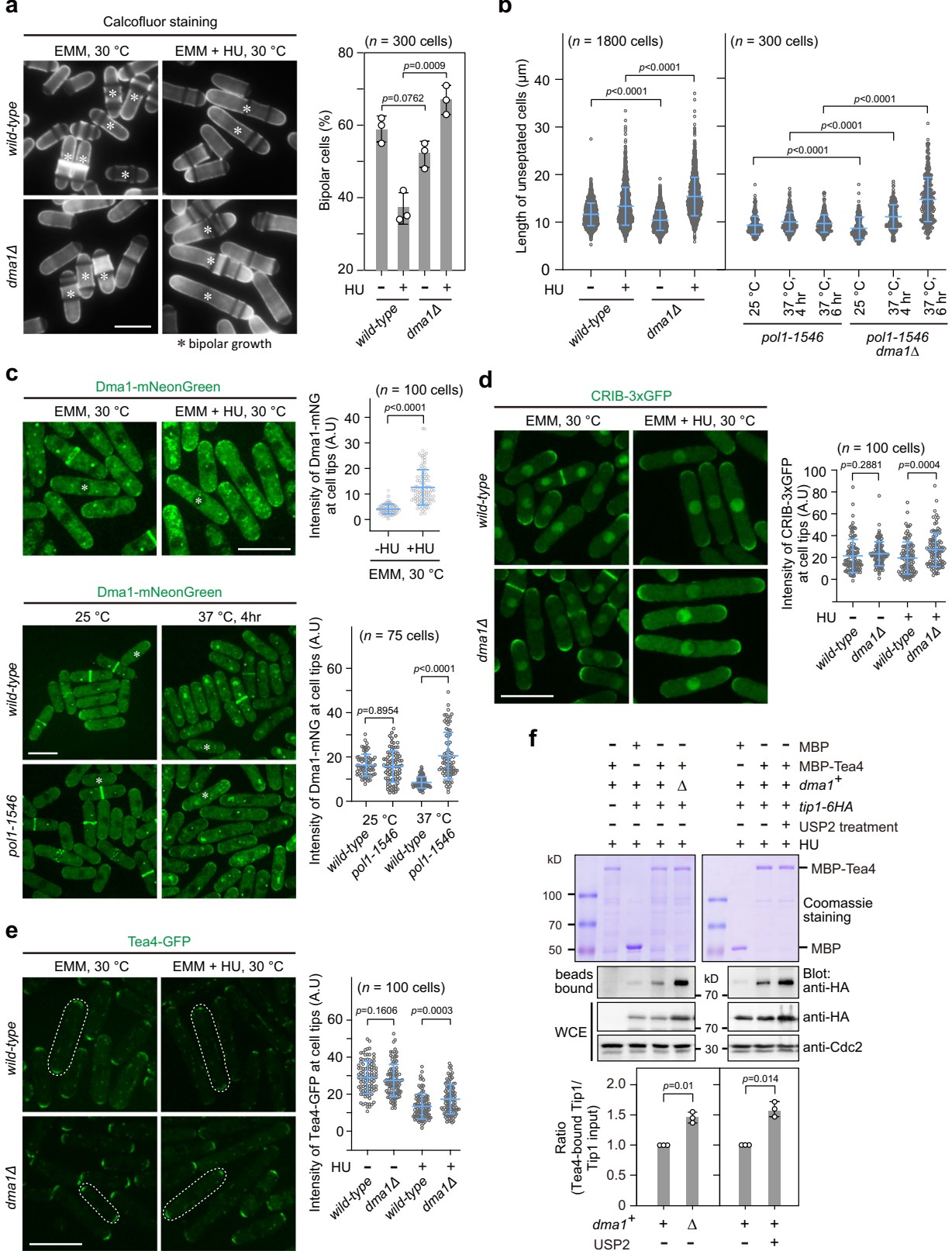

## Methods

### Yeast methods

*S. pombe* strains used and created in this study are listed in Supplementary Table 1. Liquid cultures or solid agar plates consisting of rich medium (YE5S) or synthetic minimal media (EMM2) with appropriate supplements were used, and genetic crosses and general yeast techniques were performed as described previously[48,49]. Adenine, histidine, leucine, uracil, and lysine hydrochloride were added in YE5S or EMM2 media at a final concentration at 75 mg/L in cell length measurement assays. Fission yeast cells were transformed using a lithium acetate-based procedure[50]. Expression of NTAP-Dma1, NTAP-Dma1$^{\Delta RF}$, or NTAP-Dma1-5xGly-ubiquitin$^{(7K \rightarrow 7R)}$ was controlled by the thiamine-

**Fig. 4 | Dma1 is required to restrain active bipolar growth upon defective DNA replication. a** Defective inhibition of bipolar growth in HU-treated *dma1Δ*. Cells of indicated genotypes grown in EMM5S liquid medium at 30 °C before or after being treated with 12 mM HU for 5.5 h were stained with calcofluor before scoring. Asterisks indicate bipolar cells. **b** Cell length is increased in *dma1Δ* cells upon defective DNA replication. Cells were either grown and treated with HU as in (**a**) or were cultured at 25 °C or shifted from 25 to 37 °C for 4 or 6 h (for *pol1-1546* cells). Live cell length was measured. **c** DNA replication defects induced by HU or *pol1-1546* mutation promote the accumulation of Dma1 at cell ends. Cells were grown and treated as in (**b**), cell images were captured after being fixed and fluorescence intensities of Dma1-mNeonGreen at cell ends were quantified. Asterisks indicate the representative cells enlarged in Supplementary Fig. 8b. **d**, **e** *dma1Δ* is defective in inhibition of accumulation of cell-polarity landmarks CRIB and Tea4 upon defective DNA replication induced by HU. Cells of indicated genotypes were grown, treated, and imaged as in (**c**). Fluorescence intensities of GFP signals at cell ends were quantified. Representative cells are outlined and enlarged in Supplementary Fig. 10a. **f** Binding affinity between Tea4 and Tip1 is elevated in *dma1Δ* cells and in wild-type cells treated with deubiquitinating enzyme USP2. Yeast lysates from indicated strains were treated with HU together with or without USP2 and incubated with bacteria-expressed MBP-Tea4 in *in vitro* pull-down assays. The relative levels of Tea4-bound Tip1 were quantified by comparing ratio of band intensities of Tip1-6HA from bead-bound and total lysate (WCE) with ratio in wild-type strain or USP2-untreated sample being set as 1.00. *n* = 3. Bars represent mean ± SD. For **a**–**e**, pooled data from three independent experiments are shown as mean ± SD; *n* indicates total cell numbers counted or measured for each strain; for **a**–**f**, *p* values were determined by two-tailed unpaired *t*-test. A.U. arbitrary units. Scale bars, 10 µm.

repressible *nmt41* promoter of pREP41. Expression of wild-type or mutant 6xHis-myc-ubiquitin was controlled by the thiamine-repressible *nmt1* promoter of pREP1. Expression from these *nmt* promoters was kept off by the addition of 5 mg/mL thiamine to the medium, and expression was induced by washing and culturing in medium lacking thiamine for 12–24 h. G418 disulphate (Sigma-Aldrich), hygromycin B (Sangon Biotech), and Phloxine B (Sigma-Aldrich, P4030) were added in solid YE5S plates to generate final concentrations of 100 or 5 µg/mL, respectively, where appropriate. 5-FOA (5-fluoroorotic acid) (Fermentas) was added in solid YE5S plates to get a final concentration of 0.2% for counterselection of *ura4⁺*. HU (hydroxyurea) (Sangon Biotech), ABA (Sangon Biotech), and CHX (Sigma-Aldrich) were added in liquid cultures at final concentrations of 12 mM (for HU), 250 µM (for ABA) or 100 µg/mL (for CHX) respectively. For *nda3-KM311* arrests, cultures were shifted to 18 °C for 8 h before harvesting. For *cdc25-22* or *mts3-1* arrests, cultures were shifted to 37 °C for 4 h before analysis.

## Plasmid and yeast strain construction

To construct expression vector pREP41-NTAP-dma1^ΔRF^, the fragment containing *dma1^ΔRF^* (1–573 bp of *dma1⁺* coding sequence) was amplified by PCR using yeast genomic DNA as template and primers containing *Nde*I and *Bam*HI sites, and then PCR products were digested with *Nde*I and *Bam*HI and cloned into a similarly digested pREP41-NTAP vector[51].

To construct expression vector pREP41-NTAP-dma1-5xGly-ubiquitin^(7K→7R)^, the full-length *dma1⁺* coding sequence was first amplified by PCR using yeast genomic DNA as template and primers containing *Nde*I and *Bam*HI sites and then cloned into pREP41-NTAP vector[51], this resulted in the construction of pREP41-NTAP-dma1⁺ vector. A fragment carrying ubiquitin coding sequence with all seven lysines mutated to arginines (ubiquitin^(7K→7R)^) was amplified from a plasmid (a kind gift from Ning Zheng) and cloned into pREP41-NTAP-dma1⁺ using *Bam*HI site to obtain pREP41-NTAP-dma1-ubiquitin^(7K→7R)^. Sequence of five tandem glycine codons (GGT GGA GGC GGG GGT) was then introduced between *dma1⁺* and mutated ubiquitin sequences via site-directed mutagenesis to finally get pREP41-NTAP-dma1-5xGly-ubiquitin^(7K→7R)^.

To generate the vectors for recombinant fusion protein production of MBP-Dma1(full-length), MBP-Dma1(1-173aa), MBP-Dma1(167-267aa), MBP-Dma1(R64A), and MBP-Tea4, a gene fragment was amplified by PCR using plasmids carrying the genes[22,23] or directly yeast genomic DNA as templates and then inserted into pMAL-2c (New England BioLabs, Ipswich, MA) or pET28a(+) (Novagen) as described previously[22]. Integrity of cloned DNA was verified by sequencing analysis.

Carboxy-terminal mNeonGreen epitope tagging of endogenous Dma1 was done by PCR-based gene targeting[52].

To construct the strains expressing FHA domain (1-173aa) of Dma1 fused with GFP, a *dma1⁺* gene fragment corresponding to 1-173aa was amplified by PCR using plasmids carrying the genes[22,23] as templates and then inserted into pHBKA11-CFP (a kind gift from Yoshinori

Watanabe) using *Sal*I and *Bam*HI sites. The CFP in the resultant vector pHBKA11-dma1^(1-173aa)^-CFP was replaced by GFP using the "T-type" enzyme-free cloning method[53] to get the vector pHBKA11-dma1^(1-173aa)^-GFP. This vector was then linearized by *Apa*I and integrated ectopically at *lys1⁺* locus using the *hyg^R^* marker after transformation.

To make the Dma1-CFP, CFP-Mod5, and chimeric Dma1-CFP-Mod5 fusions under the control of $P_{adh21}$ promoter, the vector pHBKA21-CFP was first generated by mutating the promoter from $P_{adh11}$ to $P_{adh21}$ using pHBKA11-CFP as a starting template via site-directed mutagenesis as described previously[54]. A short sequence containing restriction cutting sites *Kpn*I, *Nhe*I, and *Xma*I was introduced into the vector pHBKA21-CFP right behind the CFP sequence using Quikgene site-directed mutagenesis[55]. Then full-length *dma1⁺* or *mod5⁺* fragment was cloned into this modified pHBKA21-CFP vector using *Sal*I/*Bam*HI sites or *Kpn*I/*Xma*I sites respectively. Point mutations of FHA domain (R64A) or RF domain (I194A) were introduced into these vectors as desired using Quikgene site-directed mutagenesis[55]. These procedures resulted in the construction of a series of CFP-fusion vectors including pHBKA21-Dma1-CFP, pHBKA21-Dma1^R64A^-CFP, pHBKA21-Dma1^I194A^-CFP, pHBKA21-CFP-Mod5, pHBKA21-Dma1-CFP-Mod5, pHBKA21-Dma1^R64A^-CFP-Mod5, pHBKA21-Dma1^I194A^-CFP-Mod5, pHBKA21-Dma1^R64A;I194A^-CFP-Mod5. All these vectors were finally linearized by *Apa*I and integrated ectopically at *lys1⁺* locus in *dma1Δ::ura4⁺* background using the *hyg^R^* marker after transformation.

To make strains expressing full-length or truncations of *tip1* with C-terminal 6HA tag from a so-called "Z locus" (named after its neighboring *zfs1⁺* gene) on chromosome 2[56], a vector (pUC119-$P_{adh11}$-6HA(C)-hphMX6-lys1*) was first constructed by switching the GBP-mCherry fragment in vector pUC119-$P_{adh11}$-GBP-mCherry(C)-hphMX6-lys1*[54] to 6HA derived from pFA6a-6HA-kanMX4[57] using the "T-type" enzyme-free cloning method[53]. A fragment from a *Z* locus adjacent to the *zfs1⁺* gene together with kanMX6 cassette was released from a vector pKANZA21-CFP-TEV[56] cut with *Bgl*II and *Eco*RI and subcloned into pUC119-$P_{adh11}$-6HA(C)-hphMX6-lys1* to replace *hphMX6-lys1*. This generated a vector pUC119-$P_{adh11}$-6HA(C)-kanMX6-Z which carried a *Z* locus-adjacent sequence. Then sequences corresponding to full-length or desired truncations of *tip1* were cloned into the above vector by "T-type" enzyme-free cloning method[53], resulting in a series of vectors including pUC119-$P_{adh11}$-tip1^(full length)^-6HA-kanMX6-Z, pUC119-$P_{adh11}$-tip1^(1-134aa)^-6HA-kanMX6-Z, pUC119-$P_{adh11}$-tip1^(1-244aa)^-6HA-kanMX6-Z, pUC119-$P_{adh11}$-tip1^(135-244aa)^-6HA-kanMX6-Z, pUC119-$P_{adh11}$-tip1^(135-461aa)^-6HA-kanMX6-Z, pUC119-$P_{adh11}$-tip1^(245-314aa)^-6HA-kanMX6-Z and pUC119-$P_{adh11}$-tip1^(315-461aa)^-6HA-kanMX6-Z.

To make strains expressing full-length *tip1* with C-terminal fusions of DUB module of herpes simplex virus 1 UL36 protein, the sequences corresponding to catalytically active or dead versions of UL36 (residues 15-260) were amplified from plasmids pDUAL-$P_{nmt41}$-Nbr1-mCherry-UL36 or pDUAL-$P_{nmt41}$-Nbr1-mCherry-UL36(C40S)[34] and inserted behind 6HA tag in pUC119-$P_{adh11}$-tip1^(full length)^-6HA-kanMX6-Z by "T-type" enzyme-free cloning method[53]. This generated vectors of

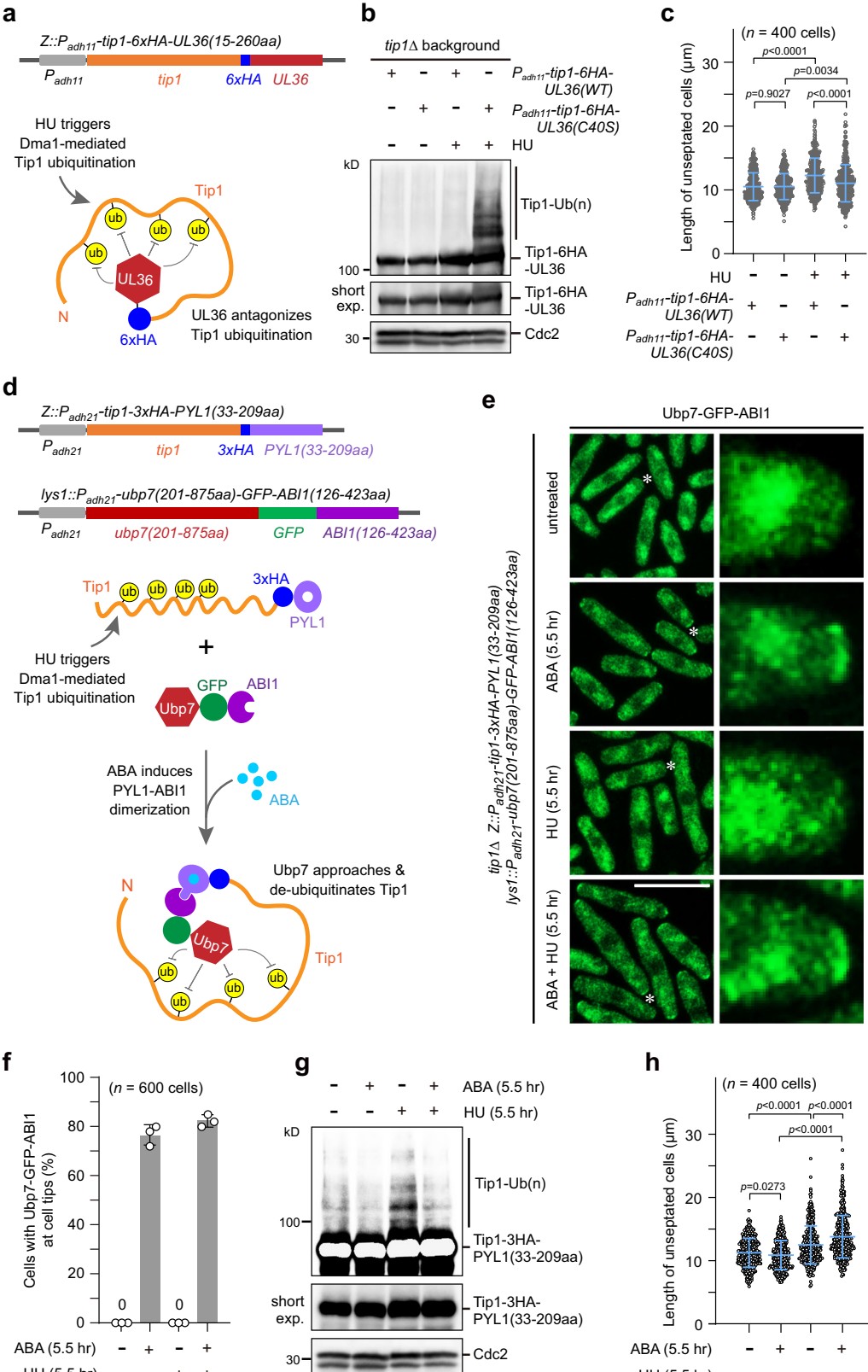

pUC119-$P_{adh11}$-tip1$^{(full\ length)}$-6HA-UL36-kanMX6-Z and pUC119-$P_{adh11}$-tip1$^{(full\ length)}$-6HA-UL36(C40S)-kanMX6-Z.

To create strains expressing full-length *tip1* with C-terminal fusion of PYL domain, the sequence corresponding to residues 33-209 of the regulatory component of ABA receptor PYL1 was amplified from *Arabidopsis thaliana* cDNA (a gift from Dr Yi Tao)

and inserted behind 3HA tag in pUC119-$P_{adh21}$-tip1$^{(full\ length)}$-3HA-kanMX6-Z by "T-type" enzyme-free cloning method[53]. To create strains expressing fusion of DUB module of Ubp7 with ABI domain, the sequences corresponding to catalytic domain (residues 201-875) of Ubp7 and ABI domain (residues 126-423) of ABI1 were amplified from yeast genomic DNA and *Arabidopsis thaliana* cDNA

**Fig. 5 | Targeted deubiquitination of Tip1 suppresses Tip1 hyper-ubiquitination and promotes cell polar growth. a–c** Fusing a deubiquitinating enzyme module of UL36 in cis with Tip1 suppresses Tip1 hyper-ubiquitination and promotes cell polar growth. **a** Schematic diagram of UL36 module (15-260aa) fusion with Tip1 ectopically expressed under the $P_{adh11}$ promoter and its potential enzymatic activity on Tip1 deubiquitination. **b, c** Cells were grown and treated with HU as in Fig. 4a and then Tip1−6HA-UL36(15-260aa) was detected by immunoblotting (**b**) and cell length was measured (**c**). The C40S mutation renders a catalytically dead version of UL36 module. **d–h** Targeted deubiquitination of Tip1 in trans based on the abscisic acid (ABA)-induced specific PYL and ABI interaction also suppresses Tip1 hyper-ubiquitination and promotes cell polar growth. **d** Scheme of the design of the Tip1-PYL and Ubp7-ABI constructs. Tip1 was fused with 3HA and the PYL domain (residues 33-209) of the ABA receptor regulatory component PYL1, and the catalytic domain (residues 201-875) of fission yeast deubiquitinating enzyme Ubp7 was fused with GFP and the ABI domain (residues 126-423) of ABI1. ABA-induced PYL-ABI dimerization may enable Ubp7 to approach and deubiquitinate Tip1. ub, ubiquitin. **e, f** Ubp7-GFP-ABI fusion protein is sufficiently recruited to cell tips by Tip1-PYL in the presence of ABA. Both Tip1−3HA-PYL and Ubp7-GFP-ABI were ectopically expressed in tip1Δ strain. Cells were grown as in Fig. 4a, then treated with 12 mM HU or 250 μM ABA separately or simultaneously for 5.5 h. Images of fixed cells were captured (**e**) and the percentages of cells with Ubp7-GFP-ABI signals at cell ends were quantified (**f**). **g, h** Ubp7(201-875aa)-GFP-ABI1(126-423aa) suppresses Tip1−3xHA-PYL1(33-209aa) hyper-ubiquitination and promotes cell polar growth. Cells were grown and treated as in (**e**) and then Tip1-3xHA-PYL1(33-209aa) was detected by immunoblotting (**g**) and cell length was measured (**h**). Asterisks indicate cell tips enlarged in insets. For **c, f,** and **h**, pooled data from three independent experiments are shown as mean ± SD; n indicates total cell numbers counted or measured for each strain; p values were determined by two-tailed unpaired t-test. Scale bar, 10 μm.

(a gift from Dr Yi Tao), respectively, and inserted in front of and behind GFP tag in pHBKA21-GFP by "T-type" enzyme-free cloning method[53]. These procedures generated vectors of pUC119-$P_{adh21}$-tip1$^{(full\ length)}$-3HA-PYL1(33-209aa)-kanMX6-Z and, pHBKA21-ubp7(201-875aa)-GFP-ABI1(126-423aa). Point mutation C217S was introduced into pHBKA21-ubp7(201-875aa)-GFP-ABI1(126-423aa) using Quikgene site-directed mutagenesis[55] to generate pHBKA21-ubp7(201-875aa, C217S)-GFP-ABI1(126-423aa).

All of these vectors were linearized by ApaI digestion and integrated at Z or lys1 locus using the $kan^R$ or $hyg^R$ marker after transformation.

Wild-type and mutants of ubiquitin fused with both 6His and myc (i.e., 6xHis-myc-ubiquitin) were expressed under the control of nmt1 promoter ($P_{nmt1}$) at the leu1$^+$ locus. To make these strains, a pREP-6His-myc-ubiquitin vector was originally constructed in which a sequence corresponding to myc (AACGGTGAACAAAAGCTAATCTCC GAGGAAGACTTGGGATCC) was inserted into pREP1-6His-ubiquitin (a kind gift from Kathy Gould) behind 6His using Quikgene site-directed mutagenesis[55]. Then, the PstI-SacI fragment containing nmt1 promoter ($P_{nmt1}$), 6His-myc-ubiquitin and nmt terminator ($T_{nmt}$) sequences from the above vector was cloned into the pJK148 vector[50] after being similarly digested with PstI and SacI, this resulted in the construction of the vector pJK148-$P_{nmt1}$-6His-myc-ubiquitin::leu1$^+$. Ubiquitin mutations in which all Lys residues except K48 or K63 mutated to Arg (K48 and K63 respectively), or all Lys residues mutated to Arg (K0) were then introduced via site-directed mutagenesis, this generated a series of vectors pJK148-$P_{nmt1}$-6His-myc-ubiquitin(K48)::leu1$^+$, pJK148-$P_{nmt1}$-6His-myc-ubiquitin(K63)::leu1$^+$ and pJK148-$P_{nmt1}$-6His-myc-ubiquitin(K0)::leu1$^+$.

To construct yeast strains carrying genomically integrated $P_{nmt41}$-NTAP-dma1$^+$ and $P_{nmt41}$-NTAP-dma1-5xGly-ubiquitin$^{(7K→7R)}$, the inserts from vectors pREP41-NTAP-dma1$^+$ and pREP41-NTAP-dma1-5xGly-ubiquitin$^{(7K→7R)}$ were cloned into pJK148-based vectors using "T-type" enzyme-free cloning method as previously described[53,54]. This generated pJK148-$P_{nmt41}$-NTAP-dma1$^+$::leu1$^+$ and pJK148-$P_{nmt41}$-NTAP-dma1-5xGly-ubiquitin$^{(7K→7R)}$::leu1$^+$.

All these pJK148-based vectors were finally linearized by NruI and integrated at leu1$^+$ locus after transformation.

All yeast transformation was performed using the lithium acetate method[50,58]. Transformants were selected on YE5S plates containing G418 or hygromycin B (for pFA6a- or pUC119-based vectors) or 0.2% 5-FOA (for replacement of ura4$^+$), or EMM2 plates lacking leucine (for pREP1/41- or pJK148-based vectors) or uracil (for pBluescript SK-ura4$^+$-based vectors). Correct tagging and integration was verified by using a colony PCR method as described previously[54] or western blotting.

DNA oligonucleotides/primers used for construction of plasmids in this study were synthesized by GENEWIZ, Inc. or Sangon Biotech, and their sequences are shown in Supplementary Table 2.

## Purification of TAP-Dma1 complexes and analysis by mass spectrometry

To prepare TAP complexes for mass spectrometry analyses, we purified NTAP-Dma1-5xGly-ubiquitin$^{(7K→7R)}$ or NTAP-Dma1$^{ΔRF}$ from 8- to 10-L cultures as previously described[22,59]. Briefly, yeast cells were grown at 32 °C overnight or first grown at 25 °C overnight, then shifted to 37 °C for 4 h (for mts3-1 strains) before being collected and lysed with a bead-beater in NP-40 buffer (6 mM $Na_2HPO_4$, 4 mM $NaH_2PO_4$, 1% NP-40, 150 mM NaCl, 2 mM EDTA, 50 mM NaF, 0.1 mM $Na_3VO_4$) containing protease inhibitor cocktail (Roche, Cat. No. 04693132001). Following the first affinity purification step on 500 μL IgG Sepharose beads (GE Healthcare; 17-0969-01) in NP-40 buffer, the cleavage reaction with 300 U TEV (Invitrogen, Cat. No. 10127-017) added was performed at 16 °C for 2 h. After the second affinity purification with 300 μL calmodulin resin (Stratagene, Cat. No. 214303-52), a single 1 mL aliquot of 20 mM EGTA-containing buffer was applied for elution. Eluates were precipitated by 25% TCA, followed by washing with cold acetone.

Purifications of NTAP-Dma1-5xGly-ubiquitin$^{(7K→7R)}$ complexes from dma1Δ and mts3-1 dnt1Δ dma1Δ backgrounds respectively, and NTAP-Dma1$^{ΔRF}$ complexes from mts3-1 dnt1Δ dma1Δ background, were each conducted one time and consequently subjected to mass spectrometry analysis. For mass spectrometry analysis, proteins were digested and analyzed by two-dimensional liquid chromatography tandem MS as previously described[60]. Briefly, proteins were resuspended in 100 mM $NH_4HCO_3$ and 8 M urea, reduced and alkylated with Tris (2-carboxyethyl) phosphine and iodoacetamide, and digested with sequencing-grade trypsin (Promega, V5111) after decreasing to 2 M urea. Peptides were loaded onto columns with a pressure cell and were separated and analyzed by three-phase multidimensional protein identification technology on a linear trap quadrupole instrument (Thermo Electron). An autosampler (FAMOS) was used for 12 salt elution steps, each with 2 μL ammonium acetate. Each injection was followed by elution of peptides with a 0–40% acetonitrile gradient except the first and last injections, in which a 0–90% acetonitrile gradient was used. Eluted ions were analyzed by one full precursor MS scan (400–2000 mass-to-charge ratio) and four tandem MS scans of the most abundant ions detected in the precursor MS scan under dynamic exclusion. MS spectra were extracted from raw files and converted to DTA files using Scansifter software (version 2.1.25). Spectra with less than 20 peaks were excluded and the remaining spectra were searched using the SEQUEST algorithm (TurboSequest, version 2.7) against a target-decoy (reversed) version of the S. pombe proteome sequence database (UniProt; downloaded 9/2010; 9.954 entries of forward and reverse protein sequences) with a precursor mass tolerance of 2.5 Da and a fragment ion mass tolerance of 0.00 Da (because of rounding in SEQUEST, this results in 0.5 Da tolerance). Peptide identifications were assembled and filtered in DTASelect software (version 1.9) using the

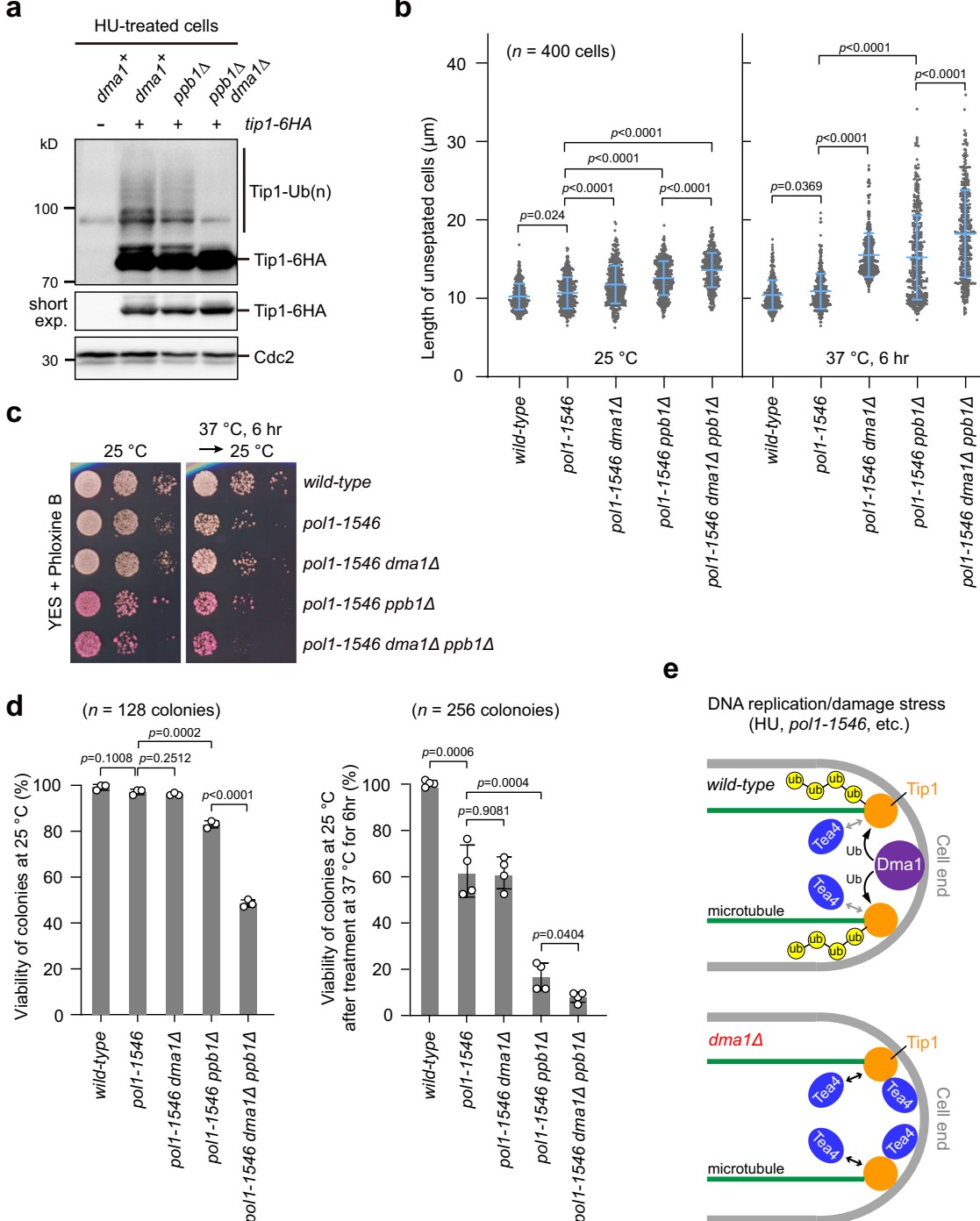

**Fig. 6 | Dma1 functions in parallel with phosphatase calcineurin in inhibition of polar growth in the presence of DNA replication stress. a** Calcineurin is not involved in Tip1 ubiquitination upon DNA replication stress induced by HU. Modification of ubiquitinated Tip1 was detected in indicated strains. **b** Additive effect of *dma1Δ* and *ppb1Δ* on cell length upon defective DNA replication induced by *pol1-1546* mutation. Cells with indicated genotypes were either cultured at 25 °C or shifted from 25 to 37 °C for 6 h. Live images were captured and cell length of unseptated cells was measured. Pooled data from three independent experiments are shown as mean ± SD; *n* indicates total cell numbers measured for each strain; *p* values were determined by two-tailed unpaired *t*-test. **c, d** Viability of colonies challenged with DNA replication stress. Yeast strains with indicated genotypes were

first grown as in (**b**), then spotted onto YES plates containing Phloxine B (**c**), or normal-looking cells of each strain were picked and placed on YES plates using a micromanipulator. Plates were incubated at 25 °C for >3 days. The frequency of viable colonies was calculated based on pooled data from 3 (for 25 °C samples) or 4 (for 37 °C-treated samples) biologically independent replicates (**d**). Pooled data are shown as mean ± SD; *p* values were determined by two-tailed unpaired *t*-test. Phloxine B was used to stain dead cells. **e** Model for Dma1-dependent signaling pathway to suspend polar growth upon DNA replication or DNA damage stress. At cell ends, ubiquitination of Tip1 by Dma1 predominantly through K63 linkages impedes polar factors (such as Tea4) to accumulate at cell ends and restrain polar growth when DNA replication or damage checkpoints are activated.

following criteria: minimum of 99.9% protein identification probability; minimum of 2 unique peptides; minimum of 95% peptide identification probability; minimum peptide length of five amino acids; minimum number of one tryptic terminus; minimum DeltaCN 0.01 or 0.08; and Xcorr minimum threshold values of 1.6, 2.3, 3.5, and 4.5 for charge states 1, 2, 3, and 4, respectively. All proteins meeting the above criteria were sorted and exported as HTML text documents.

## Immunoblotting, immunoprecipitation, and antibodies

For immunoprecipitation and western blot experiments, cells were lysed by bead disruption using FastPrep24 homogenizer (MP Biomedical) and whole-cell lysates were prepared in NP-40 buffer (6 mM $Na_2HPO_4$, 4 mM $NaH_2PO_4$, 1% NP-40, 150 mM NaCl, 2 mM EDTA, 50 mM NaF, 0.1 mM $Na_3VO_4$). To remove the potential ubiquitin or phosphorylation modifications of target proteins, cell lysates were incubated with recombinant human USP2 (R&D Systems, Cat. No. E-504) at a final concentration 50 nM for 2 h at 4 °C or 15 U of CIAP (Thermo-Fisher Scientific, Cat. No. 18009027) for 45 min at 37 °C, respectively, before being proceeded for immunoprecipitation. Proteins were immunoprecipitated by IgG Sepharose beads (GE Healthcare; 17-0969-01) (for NTAP-Dma1), mouse monoclonal anti-GFP (clone 7.1/13.1; Roche, Cat. No. 11814460001) or rat monoclonal anti-HA (clone 3F10; Roche, Cat. No. 11867423001) antibodies. Immunoblot analysis of cell lysates and immunoprecipitates was performed using peroxidase–anti-peroxidase soluble complex (Sigma-Aldrich; P1291) (1:800), rabbit polyclonal anti-Myc (GeneScript, A00172-40) (1:1000), mouse monoclonal anti-GFP (clone ME11; Beijing Ray Antibody Biotech, RM1008) (1:1000), rat monoclonal anti-HA (clone 3F10; Roche, Cat. No. 11867423001) (1:1000) or mouse monoclonal anti-Cig2 (clone 3A11/5; Santa Cruz Biotechnology, sc-53223) (1:1000) primary antibodies. Cdc2 was detected using rabbit polyclonal anti-PSTAIRE (Santa Cruz Biotechnology, sc-53) (1:1000 dilution) as loading controls. Secondary antibodies used were goat anti-mouse or goat anti-rabbit polyclonal IgG (H+L) HRP conjugates (ThermoFisher Scientific; #31430 or #32460) (1:5000–10,000). Information on the commercially available antibodies used in this study is also provided in Supplementary Table 3.

## In vitro binding and deubiquitination assay

The in vitro binding assays were performed as previously described[22]. Briefly, *tip1-6HA* cells were lysed and whole-cell lysates were prepared in NP-40 buffer as above. To remove the ubiquitin modifications of Tip1, cell lysates were incubated with recombinant human USP2 (R&D Systems, Cat. No. E-504) at a final concentration 50 nM for 2 h at 4 °C prior to in vitro binding. All bacterially produced recombinant MBP-fusion proteins of full-length or truncated Dma1 or Tea4 were expressed in *Escherichia coli* BL21(DE3) cells and purified on amylose resin (MBP; New England BioLabs) according to the manufacturer's instructions. Purified MBP-Dma1 or MBP-Tea4 proteins were incubated with clarified whole yeast-cell lysates (treated with or without USP2) for 1–2 h at 4 °C, followed by SDS-PAGE, Coomassie blue staining or western blot analysis to examine the association between full-length or truncated Dma1 and Tip1.

## In vivo Tip1 ubiquitination assay

The in vivo Tip1 ubiquitination assay was performed as previously described[61], except using the vector pREP1-6His-myc-ubiquitin instead of pREP1-6His-ubiquitin for expressing ubiquitin-fusion proteins. Briefly, Tip1-6HA strains transformed with pREP1-6His-myc-ubiquitin were grown in the absence of thiamine for 20–24 h to induce 6His-myc-ubiquitin expression. Samples were collected and extracts were prepared in lysate buffer (8 M urea, 300 mM NaCl, 50 mM $Na_3PO_4$, 0.5% NP-40, 4 mM imidazole). Ubiquitin conjugates were purified on Ni-NTA beads and washed by washing buffer (8 M urea, 300 mM NaCl,

50 mM $Na_3PO_4$, 0.5% NP-40, 20 mM imidazole) for four times, separated on SDS-PAGE, and immunoblotted with rabbit polyclonal anti-Myc (GeneScript, A00172-40) (1:1000) to detect ubiquitin and rat monoclonal anti-HA (clone 3F10; Roche, Cat. No. 11867423001) (1:1000) to detect ubiquitinated Tip1-6HA.

## Flow cytometry analyses

DNA content was analyzed with fluorescence-activated cell sorting (FACS). For the FACS analysis, the cells were first fixed with cold 70% ethanol and kept on ice for >30 min before pelleting at 12,000 rpm. Cells were then resuspended in 50 mM sodium citrate with 10 μg/mL RNase A for over 3 h. DNA was then stained with 2 μg/mL propidium iodide (PI) (Beyotime Biotechnology, ST511) before brief sonication. At least 20,000 cells were acquired per sample on a Quanteon flow cytometer (ACEA Biosciences) for PI detection (FL2-A). DNA content is shown on a linear scale after gating for single cells in FlowJo (version 10) software.

## Microscopy methods

To determine cell length at division, the yeast strains were grown in EMM5S medium with or without HU treatment to an $A_{600}$ of <0.5, and cells without clear septa under DIC microscopy were selected for measurement. A minimum of 80 unseptated cells were scored for each strain.

GFP-, CFP-, mNeonGreen-, RFP-, or tdTomato-fusion proteins were observed in live or cold methanol-fixed cells. For calcofluor staining, cells were fixed with 3.7% formaldehyde followed by staining with 5–10 μg/mL calcofluor (Sigma-Aldrich). Photomicrographs were obtained using a Nikon 80i fluorescence microscope or a Perkin Elmer spinning-disk confocal microscope (UltraVIEW VoX) with a 100× NA 1.49 TIRF oil immersion objective (Nikon) coupled to a cooled CCD camera (9100-50 EMCCD; Hamamatsu Photonics) and spinning disk head (CSU-X1, Yokogawa). Signals of Dma1-mNeonGreen, Dma1-CFP fusions, Tip1-tdTomato, CRIB-3xGFP, Tea4-GFP, and Ubp7(201-875aa, WT or C217S)-GFP-ABI1(126-423aa) were defined as being localized at cell tips when they appeared as smooth or punctate foci in a 4-μm crescent region at cortex of cell end. Fluorescence intensities of Tip1-tdTomato, Dma1-mNeonGreen, CRIB-3xGFP, or Tea4-GFP were measured along a 4-μm-long line on the cell tip on maximum projection images and the intensity ratio was calculated after background deduction. Image processing and analysis was performed using Element software (Nikon, version 2.34), Metamorph software (Molecular Devices, version 7.1), iQ software (Andor, version 3.6), ImageJ software (National Institutes of Health, versions 2.0.0-rc-69), and Adobe Photoshop (Adobe Systems, version cs6).

## Statistical analyses and reproducibility

No statistical methods were used to predetermine the sample size. The experiments were randomized and the investigators were not blinded to allocation during experiments and outcome assessment. All experiments were independently repeated two to four times with similar results obtained. For quantitative analyses of protein level, cell polar growth pattern, fluorescence intensity, and cell length of each experiment, at least 75 cells were counted or measured for each time point or sample. No data were excluded from our studies. Data collection and statistical analyses were performed using Microsoft Office Excel (Microsoft, version 2017) or GraphPad Prism (GraphPad Software Inc., version 6.0) software. Each individual measurement of fluorescence intensity or cell length is represented as a circle in the graphs. Statistical data are expressed as mean values with error bars representing ±standard deviation and were compared using two-tailed unpaired Student's $t$-tests unless indicated otherwise. Differences were considered to be statistically significant at $p < 0.05$.

**Reporting summary**

Further information on research design is available in the Nature Research Reporting Summary linked to this article.

## Data availability

The mass spectrometry raw data are not available anymore. The list of potential Dma1-interacting proteins identified by mass spectrometry in this study is provided as Supplementary Data 1. All remaining data supporting the findings of this study are available within the article and its supplementary information files. Source Data are provided with this paper.

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

## Acknowledgements
We thank Li-lin Du, Kathy Gould, Yoshinori Watanabe, Iain Hagan, Daniel P. Mulvihill, Fred Chang, Damian Brunner, Kazunori Kume, Juan Carlos G Cortés and Ning Zheng for kindly providing the yeast strains or plasmids, and Yi Tao for *Arabidopsis thaliana* cDNA. We also thank Qing-feng Liu and Xin Chen for their kind help with the confocal microscope imaging. This work was supported by the National Natural Science Foundation of China [grant numbers 31371360, 31171298, and 31671411 to Q.J.].

## Author contributions
X.W. and F.Z. performed all microscopy experiments with the help of L.H. X.W., Y.Y., and G.W. performed all immunoblotting, co-immunoprecipitation experiments, and in vitro binding assays. G.W. first observed and confirmed that Dma1 is involved in Tip1 ubiquitination. Q.J. and Y.W. conceived the study. D.M., C.F., Y.W., and Q.J. designed the experiments and supervised the research. Y.W. and Q.J. wrote the paper with the inputs from F.Z., D.M., and C.F.

## Competing interests
The authors declare no competing interests.
