## [Peer Review File · Nature Communications]

Ubiquitination of CLIP-170 family protein restrains polarized growth upon DNA replication stressREVIEWER COMMENTS

Reviewer #1 (Remarks to the Author):

Wang et al, Ubiquitination of CLIP-170 family protein restrains polarized growth upon DNA replication stress.

The authors showed, through biochemical and imaging experiments of mutated fission yeast strains, that Dma1 ubiquitinates Tip1; and that Tip1-ubiquitination inhibits cell growth when DNA replication stress is induced.

This reviewer finds the Dma1-Tip1 localization and ubiquitination data very solid. However, the conclusion that Dma1-dependent Tip1-ubiquitination blocks cell growth in response to DNA replication stress appears to be premature. While the authors did show that DNA replication stressed cells are longer in the absence of Dma1, and that ubiquitinated-Tip1 keeps cell short under DNA replication stress, they did not perform a key control experiment: What is cell length under DNA replication stress when Tip1 is absent?

This reviewer does not recommend publication, until essential control experiments have been performed.

Reviewer #2 (Remarks to the Author):

All cells control growth and polarity in response to cell cycle progression and environmental stress. In rod-shaped fission yeast cells, spatially and temporally regulated polarity factors promote cell growth at cell tips during interphase. Many of these factors, including the CLIP-170 homologue Tip1, are evolutionarily conserved. In this paper, Wang et al. show that when cells encounter DNA damage or stress during S phase, the E3 ubiquitin ligase Dma1 ubiquitinates Tip1 to inhibit polarized growth. Because CLIP-170 has been implicated in the pathogenesis of cancer, this work could provide fundamental insights behind the regulatory function of PTMs in cell polarity pathways that could drive clinical applications. The authors start by observing localization of Dma1 to growing cell tips and use a combination of biochemistry, genetics, and microscopy to trace this localization to regulation of Tip1. They employ multiple cell cycle mutants and synthetic targeting approaches to define the timing and function of this Dma1-Tip1 interaction. The authors largely support their claims with rigorous data, although further experiments could clarify several points as listed below. Overall, this paper provides a new, mechanistic link between cell growth and cell cycle control and will appeal to a wide range of cell biologists.

Major concerns:

1. From Figure 1E, the authors claim that Dma1-mNeonGreen colocalizes with Tip1-tdTomato in T-shaped cells. This claim would be strengthened by examining a greater number of T-shaped mutants, and by showing zoomed images (both single channel and merged) for the specific areas of colocalization. This suggestion of zoomed images applies to other microscopy figures in the paper as well.
2. In addition, the authors might use a starvation/refeed experiment on *tea1Δ* cells to conclude that Dma1 regulates Tip1 at actively growing cell ends. This approach would dramatically increase the number of T-shaped cells allowing easy quantification of the phenotype.
3. The authors conclude that Dma1 ubiquitinates Tip1 primarily through K63 linkages, which serves a

nonproteolytic function. However, in Figure 1E and Figure 1F, it seems clear that Dma1 ubiquitination does promote Tip1 degradation. If the authors wish to make statements "...that Dma1 ubiquitinates Tip1 specifically at cell ends, and this modification does not target Tip1 for degradation", then additional experiments that better distinguish the functions of the K48 and K63 linkages are needed. For instance, the authors might seek to repeat their experiments with cycloheximide with their ubiquitination mutants. At a minimum, the authors should revisit their phrasing throughout the paper. There are times where they apply appropriate caution in describing these results, but others in which they inappropriately push the conclusion that the interaction is independent of degradation.

4. In Figure 4E, it seems like the total cellular Tea4-GFP signal is decreased in wild-type cells following treatment with hydroxyurea. The authors should quantify total cellular GFP levels in addition to local intensities at the cell tips. This decrease in signal may not be observed in *dma1Δ* cells (?), which might suggest that Dma1 targets Tea4 for degradation. A similar concern exists for localization of CRIB in figure 4D. Local changes in concentration at cell tips are only relevant if they are not accompanied by a change in total cellular concentration.

5. In Figure 5E, I was confused by the panels, which might be mislabeled (?). The authors claim that ABA increases the tip localization, but to my eye the tip localization is lost by ABA. Part of the confusion is the cytoplasmic cluster of Ubc7-GFP-ABI1, which I assume is not included in this quantification. The true tip localization should refer to the signal at the extreme cortex, which is lost upon treatment with ABA.

6. In Figure 5F, the authors quantified the percentage of cells with Ubp7-GFP-ABI1 at cell tips, but it is not clear how this was determined. I observe Ubp7-GFP-ABI1 at cell tips in untreated cells and in cells treated with hydroxyurea for 5.5 hours. However, the authors conclude the opposite that there was more Ubp7-GFP-ABI1 at cell tips following the addition of ABA. To better understand these findings, the authors should provide a better description of how they defined cell tips to quantify the percentage of cells.

7. The authors utilize hydroxyurea (HU), which functions as an inhibitor of DNA replication, to synchronize populations of cells in S-phase. In Figure 4A, they show that the percentage of bipolar cells increases in *dma1Δ* cells following treatment with hydroxyurea. The authors claim that "deletion of *dma1+* removed the inhibition of bipolar growth posed by HU treatment and led to increased cell length." However, it also seems possible that *dma1Δ* alters the HU-triggered cell cycle checkpoint, leading to cells that have leaked into a bipolar G2 state. This concern is amplified by the established role of Dma1 in checkpoints later in the cell cycle. The authors need to add a way to confirm that these cells are still in S-phase and have not prematurely entered into G2.

Minor concerns:

1. Several of the figures appear to be mis-stated in the text. For instance, the authors state that both Figure 1A and Figure 1B show a physical interaction between Dma1 and Tip1, but only Figure 1A does this.

2. In Figure 1E, the authors need to include a marker for microtubules if they want to claim that Tip1-tdTomato localizes at the microtubule tips.

3. The authors claim to observe faint but clearly visible punctate Dma1-mNeonGreen signal at cell tips. To the reader, these punctate structures are not as obvious so the authors may wish to provide a zoomed in version of cell tips.

4. In Figure 4F, it is not clear what cells (all of them?) were treated with hydroxyurea and it is not

stated in the figure legend. The authors should add this label so that it is clear when readers look at this figure.

Reviewer #3 (Remarks to the Author):

Wang and co-workers report an ubiquitin ligase, Dma1 in mediating cell polarity in fission yeast by ubiquitinating Tip1 in fission yeast. The observation that Dma1 being localised at cell tips in fission yeast was made several years ago by Kathy Gould's lab (<https://doi.org/10.1091/mbc.e18-04-0261>) but the functional significance of Dma1 at the cell tip was unclear. By analysing the interactome of Dma1, Wang and co-workers identified a cell polarity factor Tip1 was interacting with Dma1. They show biochemically that the interaction of Dma1 to Tip1 was mediated by the FHA domain of Dma1 to the C-terminus of Tip1. They also show that Dma1 may be involved in regulating a cellular phenomenon called end take off (NETO) by ubiquitinating Tip1 when pombe cells were exposed to DNA replication stress. As Dma1 is closely related to human checkpoint proteins CHFR and RNF8, I believe that their report would be of interest to the wider scientific community.

I find the paper to be well written and logically structured. However, while I do think that the authors have good reason to believe that their proposed model to be true, I have concerns with the quality of some of the experimental data that they presented here. I strongly urge the authors of the paper to address the major points 2, 3 and 4 that I raised. Unless these concerns are addressed, I am afraid that their experimental results do not provide conclusive evidence to support their model of Dma1 regulating NETO by ubiquitinating Tip1 during DNA replication stress.

Major Points:

1. Figure 1d, the signal for Ags1-RFP and Dma1-mNeonGreen signals are very poor. This contrasts with images of Cortes Ags1-GFP cells (<https://dx.doi.org/10.1083/jcb.201202015>) and Jones Dma1-mNeonGreen cells (<https://doi.org/10.1091/mbc.e18-04-0261>) where they show the localisation of both proteins clearly at the cell tips. Due to the poor imaging, it is difficult to conclude that Ags1-RFP and Dma1-mNeonGreen have similar localisation dynamics. As the images here are obtained via live cell imaging, it is likely that photobleaching is the cause of this issue. Perhaps they authors can retake the images while taking care to minimise UV exposure. This can be achieved by focusing the microscope using white light as the pombe cells are easily seen with white light and only exposing the cells to UV when capturing the fluorescent images.

2. Figure 5 d,e,f,g,h: The paper describes an experiment using ABA to induce UBP7 to deubiquitinate Tip1. While the results can be interpreted that ABA induced interaction between UBP7 and Tip1 is preventing the ubiquitination of Tip1, the results presented cannot exclude the possibility of the binding of UBP7 to Tip1 causes the delocalisation of Tip1 from the cell tips as seen in Figure 5e. The loss of Tip1 from the cell tips could induce a null phenotype for Tip1 which would also result the increase in the length of unseptated cells which (figure 5h, <https://doi.org/10.1038/ncb2166>). To exclude this possibility, the authors can make a Ubp7 mutant that lacks the de-ubiquitination activity to determine if inducing the interaction of mutant Ubp7 with Tip1 would recapitulate the wild type response to HU treatment.

3. Cell length measurements made in Figure 5c, 5h and 6b was done on cells with variable autotrophies. It has been established that cell length can be impacted by cells of different auxotroph requirements and can have a major impact on cell length (<https://dx.doi.org/10.1101/pdb.top079764>). Strains with identical auxotroph requirements should be used when making cell length measurements.

4. Throughout the paper, cell length is used as a proxy for monopolar and bipolar tip growth. While

cell length is informative, I strongly recommend including calcofluor staining like the one used in Figure 4a, as means to directly determine the monopolar/bipolar tip growth status of cells. I am afraid that if this is not done, the conclusion that Dma1 directly regulates polarised tip growth upon DNA replication stress via ubiquitinylation of Tip1 cannot be properly substantiated.

Minor comment:

1. Dma1-5xGly-Ub(7KR) fusion protein is not a commonly used mutant (line 98 to 102). Can a reference be used to support the statement that this mutant is self-conjugating to target substrates? Else, experimental evidence should be supplied to support this conclusion.

2. The figure referenced in Line 153 should be Fig. 2d

Our point-by-point responses to three reviewers' comments.

Please note that the reviewers' comments are italicized and our responses are in blue color.

REVIEWER COMMENTS

Reviewer #1 (Remarks to the Author):

Wang et al, Ubiquitination of CLIP-170 family protein restrains polarized growth upon DNA replication stress.

The authors showed, through biochemical and imaging experiments of mutated fission yeast strains, that Dma1 ubiquitinates Tip1; and that Tip1-ubiquitination inhibits cell growth when DNA replication stress is induced.

This reviewer finds the Dma1-Tip1 localization and ubiquitination data very solid. However, the conclusion that Dma1-dependent Tip1-ubiquitination blocks cell growth in response to DNA replication stress appears to be premature. While the authors did show that DNA replication stressed cells are longer in the absence of Dma1, and that ubiquitinated-Tip1 keeps cell short under DNA replication stress, they did not perform a key control experiment: What is cell length under DNA replication stress when Tip1 is absent?

This reviewer does not recommend publication, until essential control experiments have been performed.

Authors' Response:

Actually we performed this control experiment and found that *tip1* loss-of-function mutation does not mimic the effects of manipulation of Tip1 ubiquitination.

We examined the cell length in HU-treated *tip1Δ* and *dma1Δ tip1Δ* mutants (please see our results in author response Figure 1 below), and found that *tip1Δ* cells are generally longer than *wild-type* cells no matter HU is present or absent. Consistently, as reported in the previously published paper (Kume et al., Nature Cell Biology, 2011; <https://doi.org/10.1038/ncb2166>), deletion of *tip1*⁺ also causes *pol1-1546* cells to switch from monopolar growth to bipolar growth at restrictive temperature (Please see Fig. 1b in that paper, strain #11). This switch likely increases cell length of double mutant *pol1-1546 tip1Δ*, though they did not show any data on cell length measurements. These data suggest that cell growth pattern in *tip1* deleted cells is very distinct from that in cells when Tip1 is having or lacking modifications.

We think our observed cell length increase in *tip1* deletion mutant is most likely due to the possibility that the other key polarity factor Tea1-mediated polar growth becomes compensatory or even dominant in the absence of *tip1*⁺, because Tea1 can be targeted to the

cell ends through both Tip1-dependent and Tip1-independent mechanisms (Please see two previous reports: Brunner & Nurse, Cell. 2000; doi:10.1016/s0092-8674(00)00091-x and Snaith & Sawin, Nature, 2003; doi:10.1038/nature01672). However, we have been unable to unambiguously test whether this is indeed the case, because *tea1* deletion causes branched cell shape, which prevents us from conclusively measuring cell length of bipolar cells.

Reviewer #2 (Remarks to the Author):

All cells control growth and polarity in response to cell cycle progression and environmental stress. In rod-shaped fission yeast cells, spatially and temporally regulated polarity factors promote cell growth at cell tips during interphase. Many of these factors, including the CLIP-170 homologue Tip1, are evolutionary conserved. In this paper, Wang et al. show that when cells encounter DNA damage or stress during S phase, the E3 ubiquitin ligase Dma1 ubiquitinates Tip1 to inhibit polarized growth. Because CLIP-170 has been implicated in the pathogenesis of cancer, this work could provide fundamental insights behind the regulatory function of PTMs in cell polarity pathways that could drive clinical applications. The authors start by observing localization of Dma1 to growing cell tips and use a combination of biochemistry, genetics, and microscopy to trace this localization to regulation of Tip1. They employ multiple cell cycle mutants and synthetic targeting approaches to define the timing and function of this Dma1-Tip1 interaction. The authors largely support their claims with rigorous data, although further experiments could clarify several points as listed below. Overall, this paper provides a new, mechanistic link between cell growth and cell cycle control and will appeal to a wide range of cell biologists.

Major concerns:

1. From Figure 1E, the authors claim that *Dma1-mNeonGreen* colocalizes with *Tip1-tdTomato* in T-shaped cells. This claim would be strengthened by examining a greater number of T-shaped mutants, and by showing zoomed images (both single channel and

merged) for the specific areas of colocalization. This suggestion of zoomed images applies to other microscopy figures in the paper as well.

Authors' Response:

We thank this reviewer for this suggestion. Although we are confident in our data showing Dma1 localization with Tip1 at cell tips in T-shaped cells during growth under normal conditions, which is consistent with previous publications, we decided to remove this data (for example, the images in Fig. 1e in previous version of manuscript) and concentrate on localization of these proteins when DNA replication is blocked, which is the focus of this paper.

To follow this reviewer's suggestion, we have added zoomed images for Fig. 1e for better visualization of fluorescent signals at cell ends. We have also added zoomed images for other microscopy images either in the main text figures or in supplementary figures.

2. In addition, the authors might use a starvation/refeed experiment on *tea1Δ* cells to conclude that Dma1 regulates Tip1 at actively growing cell ends. This approach would dramatically increase the number of T-shaped cells allowing easy quantification of the phenotype.

Authors' Response:

We thank this reviewer for this technical suggestion. As we have explained above in our response to 1st comment, we decided not to complicate the readers with the role of Dma1 in vegetative cells, therefore we have deleted those images involved *tea1Δ* mutant cells.

3. The authors conclude that Dma1 ubiquitinates Tip1 primarily through K63 linkages, which serves a nonproteolytic function. However, in Figure 1E and Figure 1F, it seems clear that Dma1 ubiquitination does promote Tip1 degradation. If the authors wish to make statements "...that Dma1 ubiquitinates Tip1 specifically at cell ends, and this modification does not target Tip1 for degradation", then additional experiments that better distinguish the functions of the K48 and K63 linkages are needed. For instance, the authors might seek to repeat their experiments with cycloheximide with their ubiquitination mutants. At a minimum, the authors should revisit their phrasing throughout the paper. There are times where they apply appropriate caution in describing these results, but others in which they inappropriately push the conclusion that the interaction is independent of degradation.

Authors' Response:

We apologize for the misleading conclusion we have put in our earlier version of manuscript. We guess this reviewer is referring the Figure 2e and Figure 2f in our manuscript. Our ubiquitination assays using various version of ubiquitin mutants demonstrated that Tip1 is more prominently conjugated by K63 linkages than K48 linkages, though Dma1 is able to

catalyze ubiquitin chain formation of both linkages on Tip1 as shown in our Fig. 2g. Also, the absence of *dma1*⁺ does not affect the abundance of Tip1 at cell ends with or without HU treatment (Please see our results in Fig. 3f). These results indicate that the Tip1 at cell ends that ubiquitinated by Dma1, prominently through K63 linkages, is not targeted for degradation. We agree with the reviewer that without study to distinguish functions of K48 and K63 linkages, it is premature to conclude the non-proteolytic role of K63 linkage by Dma1 at cell ends. Our speculation is those biochemical results cannot provide spatial information as these assays were performed with cell lysate. Possibly Dma1-mediated Tip1 ubiquitination through K48 linkages, which is destined for degradation, only occurs in cytoplasm.

To avoid the overstatement, we have revised the relevant description and conclusion in our revised manuscript with caution to avoid the potential misinterpretation of our data.

4. In Figure 4E, it seems like the total cellular Tea4-GFP signal is decreased in wild-type cells following treatment with hydroxyurea. The authors should quantify total cellular GFP levels in addition to local intensities at the cell tips. This decrease in signal may not be observed in dma1Δ cells (?), which might suggest that Dma1 targets Tea4 for degradation. A similar concern exists for localization of CRIB in figure 4D. Local changes in concentration at cell tips are only relevant if they are not accompanied by a change in total cellular concentration.

Authors' Response:

We can understand the reviewer's concern. To clarify the issue on the possible degradation of Tea4 and CRIB, we have performed Western blotting to examine the total cellular levels of Tea4-GFP or CRIB-3xGFP in *wild-type* and *dma1Δ* cells before and after HU treatment. We found that Tea4-GFP or CRIB-3xGFP protein levels are comparable in *wild-type* and *dma1Δ* cells. Thus, the change with the intensities of Tea4-GFP or CRIB-3xGFP at cell ends after HU treatment in *dma1Δ* cells compared to *wild-type* cells (as shown in Fig. 4d, 4e) should be attributed to local enrichment rather than overall increased protein abundance. We have included this new data in Supplementary Fig. 9b and 10b in our revised manuscript.

5. In Figure 5E, I was confused by the panels, which might be mislabeled (?). The authors claim that ABA increases the tip localization, but to my eye the tip localization is lost by ABA. Part of the confusion is the cytoplasmic cluster of Ubc7-GFP-ABI1, which I assume is not included in this quantification. The true tip localization should refer to the signal at the extreme cortex, which is lost upon treatment with ABA.

Authors' Response:

We apologize for the mislabeling during our earlier version of figure preparation. We have corrected the mistake in our revised manuscript (see new Fig 5e in our revised manuscript).

6. In Figure 5F, the authors quantified the percentage of cells with Ubp7-GFP-ABI1 at cell tips, but it is not clear how this was determined. I observe Ubp7-GFP-ABI1 at cell tips in untreated cells and in cells treated with hydroxyurea for 5.5 hours. However, the authors conclude the opposite that there was more Ubp7-GFP-ABI1 at cell tips following the addition of ABA. To better understand these findings, the authors should provide a better description of how they defined cell tips to quantify the percentage of cells.

Authors' Response:

We apologize for the confusion caused by our mislabeling on earlier version of the figure. We have corrected the label in new Fig. 5e, and now the quantified data shown in Fig. 5f are consistent with the corresponding representative images in Fig. 5e.

We have also added a description on how we defined cell tip localization of Dma1-mNeonGreen, Dma1-CFP fusions, Tip1-tdTomato, CRIB-3xGFP, Tea4-GFP and Ubp7(201-875aa, WT or C217S)-GFP-ABI1(126-423aa) for quantifications in the "Materials and Methods" section of our revised manuscript.

7. The authors utilize hydroxyurea (HU), which functions as an inhibitor of DNA replication, to synchronize populations of cells in S-phase. In Figure 4A, they show that the percentage of bipolar cells increases in *dma1Δ* cells following treatment with hydroxyurea. The authors claim that "deletion of *dma1+* removed the inhibition of bipolar growth posed by HU treatment and led to increased cell length." However, it also seems possible that *dma1Δ* alters the HU-triggered cell cycle checkpoint, leading to cells that have leaked into a bipolar G2 state. This concern is amplified by the established role of Dma1 in checkpoints later in the cell cycle. The authors need to add a way to confirm that these cells are still in S-phase and have not prematurely entered into G2.

Authors' Response:

We can understand the reviewer's concern. To follow the reviewer's suggestion, we have employed two strategies to examine whether the HU-treated *dma1Δ* cells are still in S phase.

First, we examined DNA content of both *wild-type* and *dma1Δ* cells before and after HU treatment by flow cytometer. We included G₁ phase-arrested *cdc10-v50* and G₂ phase-arrested *cdc25-22* cells as controls. We found that both HU-treated *wild-type* and *dma1Δ* cells showed DNA content between 1C and 2C, indicating they were well arrested in the S phase (see new Supplementary Figure 4a).

As a second strategy, we examined the cellular levels of S phase cyclin Cig2 by Western blotting using anti-Cig2 antibodies. We found that both HU-treated *wild-type* and *dma1Δ* cells showed comparable levels of Cig2, which were significantly elevated compared to those in

asynchronized cells (see new Supplementary Figure 4b), confirming that *dma1Δ* cells can be arrested at S phase by HU treatment as efficiently as *wild-type* cells.

We have added the above results in Supplementary Figure 4 in our revised manuscript.

Minor concerns:

1. Several of the figures appear to be mis-stated in the text. For instance, the authors state that both Figure 1A and Figure 1B show a physical interaction between Dma1 and Tip1, but only Figure 1A does this.

Authors' Response:

We thank the reviewer for pointing this concern out. Co-immunoprecipitation and cell lysate pull-down assays are two most commonly used techniques to detect the physical interaction between two proteins. In our Figure 1a and 1c, we employed co-immunoprecipitation to examine the interaction between yeast-expressed full-length Dma1 and Tip1 or between yeast-expressed N-terminal fragment of Dma1 (1-173aa) and various truncations of Tip1 respectively. In Figure 1b, we used the bacteria-expressed Dma1 or its mutants and yeast-expressed full-length Tip1-6HA to perform *in vitro* pull-down assay to identify the Tip1-interacting domain in Dma1. Both assays allowed us to test if Dma1 and Tip1 interact with each other in either direct or indirect manner. Therefore, we think both Figure 1a and Figure 1b serve the purpose of testing the physical interaction.

2. In Figure 1E, the authors need to include a marker for microtubules if they want to claim that Tip1-tdTomato localizes at the microtubule tips.

Authors' Response:

As the major point we want to demonstrate in Figure 1e is the occasional co-localization between Dma1-mNeonGreen and Tip1-tdTomato at cell ends, therefore we did not include the microtubule marker here.

We have rewritten the description on Figure 1e to avoid confusion.

3. The authors claim to observe faint but clearly visible punctate Dma1-mNeonGreen signal at cell tips. To the reader, these punctate structures are not as obvious so the authors may wish to provide a zoomed in version of cell tips.

Authors' Response:

To follow this reviewer's suggestion, we have added zoomed images for Fig. 1e for better visualization of fluorescent signals.

4. In Figure 4F, it is not clear what cells (all of them?) were treated with hydroxyurea and it is not stated in the figure legend. The authors should add this label so that it is clear when readers look at this figure.

Authors' Response:

We thank the reviewer for pointing this out. All samples used in these experiments were treated with hydroxyurea (HU). We have added the description of HU treatment in the figure legend and also added the label in Figure 4f.

Reviewer #3 (Remarks to the Author):

Wang and co-workers report an ubiquitin ligase, Dma1 in mediating cell polarity in fission yeast by ubiquitinating Tip1 in fission yeast. The observation that Dma1 being localised at cell tips in fission yeast was made several years ago by Kathy Gould's lab (<https://doi.org/10.1091/mbc.e18-04-0261>) but the functional significance of Dma1 at the cell tip was unclear. By analysing the interactome of Dma1, Wang and co-workers identified a cell polarity factor Tip1 was interacting with Dma1. They show biochemically that the interaction of Dma1 to Tip1 was mediated by the FHA domain of Dma1 to the C-terminus of Tip1. They also show that Dma1 may be involved in regulating a cellular phenomenon called end take off (NETO) by ubiquitinating Tip1 when pombe cells were exposed to DNA replication stress. As Dma1 is closely related to human checkpoint proteins CHFR and RNF8, I believe that their report would be of interest to the wider scientific community.

I find the paper to be well written and logically structured. However, while I do think that the authors have good reason to believe that their proposed model to be true, I have concerns with the quality of some of the experimental data that they presented here. I strongly urge the authors of the paper to address the major points 2, 3 and 4 that I raised. Unless these concerns are addressed, I am afraid that their experimental results do not provide conclusive evidence to support their model of Dma1 regulating NETO by ubiquitinating Tip1 during DNA replication stress.

Major Points:

1. Figure 1d, the signal for Ags1-RFP and Dma1-mNeonGreen signals are very poor. This contrasts with images of Cortes Ags1-GFP cells (<https://dx.doi.org/10.1083/jcb.201202015>) and Jones Dma1-mNeonGreen cells (<https://doi.org/10.1091/mbc.e18-04-0261>) where they show the localisation of both proteins clearly at the cell tips. Due to the poor imaging, it is difficult to conclude that Ags1-RFP and Dma1-mNeonGreen have similar localisation dynamics. As the images here are obtained via live cell imaging, it is likely that photobleaching is the cause of this issue. Perhaps the authors can retake the images while

taking care to minimise UV exposure. This can be achieved by focusing the microscope using white light as the pombe cells are easily seen with white light and only exposing the cells to UV when capturing the fluorescent images.

Authors' Response:

As this reviewer already noticed from previous studies, both Ags1-RFP and Dma1-mNeonGreen signals are clearly localized at the cell tips in vegetatively growing cells, thus we are not the first to report their localization at cell ends. We used Ags1-RFP only as an indicator for actively growing cell ends. Based on the data we presented in our paper, we did not find any evidence to show that Dma1 is involved in cell polar growth during vegetative growth, though Dma1 is largely enriched at actively growing cell ends, so Ags1 and Dma1 do not necessarily colocalize exactly at cell ends or show similar localization dynamics.

The impression of the poor quality of images shown in Figure 1d by the reviewer was most likely due to the fact that all the figures were in PDF format, which is required by the journal for the initial manuscript submission to minimize the file size. Unfortunately, this leads to lower resolution for pictures. For revised manuscript, we will upload higher resolution figures and also have included enlarged and colour merged images of cells expressing both Dma1-mNeonGreen and Ags1-RFP in Supplementary Figure 2 for better visualization of those signals at cell ends.

2. *Figure 5 d,e,f,g,h: The paper describes an experiment using ABA to induce UBP7 to deubiquitinate Tip1. While the results can be interpreted that ABA induced interaction between UBP7 and Tip1 is preventing the ubiquitination of Tip1, the results presented cannot exclude the possibility of the binding of UBP7 to Tip1 causes the delocalisation of Tip1 from the cell tips as seen in Figure 5e. The loss of Tip1 from the cell tips could induce a null phenotype for Tip1 which would also result the increase in the length of unseptated cells which (figure 5h, <https://doi.org/10.1038/ncb2166>). To exclude this possibility, the authors can make a Ubp7 mutant that lacks the de-ubiquitination activity to determine if inducing the interaction of mutant Ubp7 with Tip1 would recapitulate the wild type response to HU treatment.*

Authors' Response:

We can understand the reviewer's concern. To follow the reviewer's suggestion, we constructed a yeast strain simultaneously expressing Tip1-3xHA-PYL1(33-209aa) and GFP-ABI fused with catalytically dead Ubp7 module harboring a C217S mutation (i.e. Ubp7^(201-875aa, C217S)-GFP-ABI^(126-423aa)). When the specific interaction between Tip1-3xHA-PYL1^(33-209aa) and Ubp7^(201-875aa, C217S)-GFP-ABI^(126-423aa) was induced by ABA, Tip1 hyper-ubiquitination is not suppressed, which is similar to *wild-type* cells, though Ubp7^(201-875aa, C217S)-GFP-ABI^(126-423aa) was able to be efficiently targeted to cell tips. Also, cell polar growth is inhibited as observed in *wild-type* cells upon HU treatment. We have added these new data in Supplementary Figure 11b-f in our revised manuscript.

3. Cell length measurements made in Figure 5c, 5h and 6b was done on cells with variable autotrophies. It has been established that cell length can be impacted by cells of different auxotroph requirements and can have a major impact on cell length (<https://dx.doi.org/10.1101/pdb.top079764>). Strains with identical auxotroph requirements should be used when making cell length measurements.

Authors' Response:

This is a valid concern. Actually, we were aware of this issue during our initial and subsequent measurements on cell length. In our experiments, we added the supplements in our YE5S or EMM5S media at a final concentration of 75 mg/L instead of routinely used 200-225 mg/L, including adenine, histidine, leucine, uracil and lysine hydrochloride. This concentration is even lower than that suggested by Janni Petersen and Paul Russell in their media recipe (Cold Spring Harb Protoc, 2016; doi:10.1101/pdb.top079764), in which supplementation with 150 mg/L of the relevant amino acid is recommended to minimize variation of cell length at division, as higher concentration of some supplements (for example leucine) may transiently blocks mitotic commitment and shortens cell length. We have added the information on used concentration of supplements in our revised manuscript in the section of "Yeast methods" in "Materials and Methods".

As for cell length measurements in Figure 5c, we used two yeast strains with the exactly same autotrophies (please see in our yeast strain list: JY10406 ($h^2 tip1\Delta::kan^R Z::P_{adh11}-tip1^{(full\ length)}-6HA-linker-UL36^{15-260aa}::kan^R leu1^- ura4^- ade6^-$) and JY10407 ($h^2 tip1\Delta::kan^R Z::P_{adh11}-tip1^{(full\ length)}-6HA-linker-UL36^{15-260aa\ (C40S)}::kan^R leu1^- ura4^- ade6^-$)), so the possible effect of different auxotroph requirements on cell length should have been eliminated.

To solve the potential concern on cell length measurements we showed in Figure 5h and 6b, we have re-constructed yeast strains with all prototrophic genotypes and repeated cell length measurements with the newly constructed strains. We have included these new results of cell length measurements in Supplementary Figure 11e and Supplementary Figure 12 in our revised manuscript. We noticed that the new cell length measurement data with same prototrophic genotype strains are almost identical to those obtained previously with auxotrophic genotype strains, confirming that the yeast media recipe we have been using do not influence the cell length measurements.

4. Throughout the paper, cell length is used as a proxy for monopolar and bipolar tip growth. While cell length is informative, I strongly recommend including calcofluor staining like the one used in Figure 4a, as means to directly determine the monopolar/bipolar tip growth status of cells. I am afraid that if this is not done, the conclusion that Dma1 directly regulates polarised tip growth upon DNA replication stress via ubiquitinylation of Tip1 cannot be properly substantiated.

Authors' Response:

We thank the reviewer for this suggestion. To include the cell polarity information in the experiments, we have performed calcofluor staining for cells used in experiments shown in Figure 4b, 5c and 5h. We have added the scored polar growth pattern data based on calcofluor staining in Supplementary Figure 7, 11a and 11f respectively in our revised manuscript. As expected, our results showed that increased cell length always accompany higher frequency of bipolar growth in *dma1Δ* cells and in cells with targeted de-ubiquitination of Tip1 when they are challenged with DNA replication stress.

Minor comment:

1. *Dma1-5xGly-Ub(7KR) fusion protein is not a commonly used mutant (line 98 to 102). Can a reference be used to support the statement that this mutant is self-conjugating to target substrates? Else, experimental evidence should be supplied to support this conclusion.*

Authors' Response:

We thank the reviewer for the suggestion. Actually, we designed this experiment not based on any established strategy, it was just our naïve idea that the Dma1-5xGly-Ub(7KR) fusion could possibly self-conjugate to target substrates, thus likely enrich its ubiquitination substrates in purified complexes.

To follow the reviewer's suggestion, we set up an experiment to test whether Dma1-5xGly-Ub(7KR) fusion indeed self-conjugates to Tip1, one of the target substrates we have identified in original purifications. We constructed two yeast strains carrying *tip1-6HA* and P_{nmt41} -NTAP-*dma1*⁺ or P_{nmt41} -NTAP-*dma1-5xGly-Ub(7KR)* respectively. The association of Tip1 to NTAP-Dma1 and NTAP-Dma1-5xGly-Ub(7KR) was assessed by immunoprecipitation of two NTAP-tagged Dma1 versions. We found that, although Dma1-5xGly-Ub(7KR) fusion does not self-conjugate to Tip1 as expected, it indeed co-immunoprecipitates significantly more Tip1 than unmodified Dma1 does for unknown reasons. Nevertheless, this explains why Tip1 was initially enriched in our series of TAP-Dma1 purifications. We have added this new result in Supplementary Figure 1c in our revised manuscript.

2. *The figure referenced in Line 153 should be Fig. 2d.*

Authors' Response:

In Line 153, we intended to refer to the Dma1 mutant (Dma(R64A)) that has a compromised Tip1 binding capability. In Figure 2d, we used Dma1(R64A) mutant to show that Tip1 ubiquitination was abolished in Dma1(R64A) mutant, which is not the direct evidence to show the binding capacity change in Dma1 mutant. Therefore, we think Figure 1b is an appropriate reference in this context.

REVIEWERS' COMMENTS

Reviewer #1 (Remarks to the Author):

In the revised version, the authors have addressed previous concerns. I am now in favor of publication.

Reviewer #2 (Remarks to the Author):

The authors have addressed my comments and have improved the manuscript with new experiments and text-based changes. The paper shows that E3 ligase Dma1 promotes ubiquitination of Tip1 with functional contributions to control of bipolar growth. My only lingering concern is the continued emphasis that Tip1 ubiquitination primarily acts through non-proteolytic regulation. I understand that K63-linked modification is not classically considered a mark for proteolytic degradation. However, the authors reproducibly measure higher Tip1 protein levels in *dma1Δ* mutants, and very nicely demonstrate that the degradation rate of Tip1 is much slower in various *dma1* mutants. Most data in the paper strongly support Dma1-mediated control of Tip protein levels as the underlying mechanism. On the other hand, I do not find strong support for ubiquitination regulating the Tip1-Tea4 physical interaction. The immunoprecipitation in panel 4f is complicated by the fact that Tip1 is much more abundant in *dma1Δ* cells compared to wild type. This increased concentration is strongly expected to drive enhanced binding of Tip1-Tea4, consistent with the result shown. I would encourage the authors to revise their text further (especially in the Discussion) in an effort to address the clear role of Dma1 in regulating Tip1 protein level and degradation kinetics.

Reviewer #3 (Remarks to the Author):

The additional experimental data that the authors provided sufficiently addresses my concerns. I find the data to be convincing in supporting their experimental model and therefore would recommend that their paper be accepted for publication.

Our point-by-point responses to three reviewers' comments.

Please note that the reviewers' comments are italicized and our responses are in blue color.

REVIEWERS' COMMENTS

Reviewer #1 (Remarks to the Author):

In the revised version, the authors have addressed previous concerns. I am now in favor of publication.

Authors' Response:

We really appreciate this reviewer's positive comment.

Reviewer #2 (Remarks to the Author):

The authors have addressed my comments and have improved the manuscript with new experiments and text-based changes. The paper shows that E3 ligase Dma1 promotes ubiquitination of Tip1 with functional contributions to control of bipolar growth. My only lingering concern is the continued emphasis that Tip1 ubiquitination primarily acts through non-proteolytic regulation. I understand that K63-linked modification is not classically considered a mark for proteolytic degradation. However, the authors reproducibly measure higher Tip1 protein levels in dma1Δ mutants, and very nicely demonstrate that the degradation rate of Tip1 is much slower in various dma1 mutants. Most data in the paper strongly support Dma1-mediated control of Tip protein levels as the underlying mechanism. On the other hand, I do not find strong support for ubiquitination regulating the Tip1-Tea4 physical interaction. The immunoprecipitation in panel 4f is complicated by the fact that Tip1 is much more abundant in dma1Δ cells compared to wild type. This increased concentration is strongly expected to drive enhanced binding of Tip1-Tea4, consistent with the result shown. I would encourage the authors to revise their text further (especially in the Discussion) in an effort to address the clear role of Dma1 in regulating Tip1 protein level and degradation kinetics.

Authors' Response:

We can understand the reviewer's concern. We agree with the reviewer that it is clear that Dma1 affects the overall levels and stability of Tip1, as shown in our Fig. 2e and 2f. However, based on two important observations (see below), we still favor the idea that Dma1-mediated Tip1 ubiquitination likely affects nonproteolytic functions of Tip1 in polar growth by inhibiting its association with Tea4 at cell ends. In addition to our finding of preponderance of K63 over K48 linkages on Tip1 (as shown in Fig. 2g), we also made two other important observations. First, the absence of dma1⁺ does not affect the abundance of Tip1 measured by its intensity at cell ends with or without HU treatment (Fig. 3f), this indicates that the Dma1-mediated ubiquitination of Tip1 at cell ends is likely not targeted for degradation. Second, deletion of tip1⁺ does not rescue the increased growth phenotype of HU-treated

dma1Δ cells (Supplementary Fig. 14), this suggests that Dma1 may be working through regulating Tip1 through K63 ubiquitination, which cannot be mimicked by deletion of Tip1. We should mention that the Supplementary Fig. 14 is our newly added figure in response to this reviewer's concern.

To follow the reviewer's suggestion and to avoid the overstatement, we have further revised the relevant conclusion and discussion in our revised manuscript with caution to avoid the potential misinterpretation of our data. Please see those changes in lines #202-205 and #300-312 in our revised manuscript.

As for the Tip1-Tea4 physical interaction, our raw data of measured total protein levels of Tip1 and Tea4-bound Tip1 levels in *dma1Δ* and wild-type cells in three duplicated experiments showed that total protein levels of Tip1 increased roughly 1.5-2.5 fold in *dma1Δ* compared to wild-type cells, but Tea4-bound Tip1 levels increased roughly 2.3-3.6 fold. Please see these raw data in the Source Data file we provided. So it is clear that the increased protein levels of Tip1 in *dma1Δ* cells are not linearly proportional to the extent of significantly enhanced binding of Tip1-Tea4. Thus, our observation of enhanced binding of Tip1-Tea4 in *dma1Δ* cells cannot be simply explained as a direct consequence of increased concentration of Tip1.

Reviewer #3 (Remarks to the Author):

The additional experimental data that the authors provided sufficiently addresses my concerns. I find the data to be convincing in supporting their experimental model and therefore would recommend that their paper be accepted for publication.

Authors' Response:

We really appreciate the reviewer's positive comment.